# ConEx: Human-Interpretable Saliency Maps via Concept-Aware Attribution

**Yehonatan Elisha** [1]   **Oren Barkan** [2]   **Ziv Weiss Haddad** [1]   **Noam Koenigstein** [1]

## Abstract

Many visual explanation methods in computer vision highlight pixel importance but struggle to link these low-level cues to semantically meaningful concepts, limiting their interpretability and trustworthiness. We introduce Concept-based Explanations (**ConEx**), a novel framework that bridges saliency visualization with concept-based reasoning to provide both faithfulness and interpretability. ConEx automatically discovers class-specific concepts and represents them through concept activation vectors (CAVs), learned without manual supervision using an architecture-specific masking mechanism that reduces noise introduced by the segmentation masks to enhance concept purity. ConEx generates faithful saliency maps that reveal where each concept appears in the image and how it contributes to the prediction. To evaluate the reliability of these learned concepts, we propose two complementary metrics, Vector-Concept Match (VCM) and Concept-Class Match (CCM), that quantify concept alignment and enable direct comparison with existing methods. Extensive experiments across diverse settings demonstrate that ConEx achieves state-of-the-art performance on faithfulness, segmentation, and concept-quality benchmarks. Overall, ConEx advances the field toward truly interpretable and concept-grounded explanations in vision models. Our code is provided at: https://github.com/yonisGit/conex

## 1. Introduction

Modern vision architectures, spanning both CNNs (Simonyan et al., 2013; He et al., 2016; Huang et al., 2017; Li et al., 2022) and Vision Transformers (Dosovitskiy et al., 2021; He et al., 2022), deliver state-of-the-art accuracy yet remain inherently opaque. As these models increas-

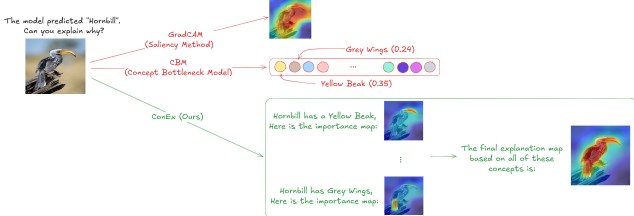

*Figure 1.* The ConEx Motivation: standard saliency maps (top) indicate where the model attends but not what it perceives, while CBMs (middle) reveal what the model detects but lack spatial grounding. ConEx (bottom) unifies both perspectives by decomposing predictions into spatially grounded, human-interpretable visual concepts (e.g., "yellow beak") and providing concept-based explanation map.

ingly inform high-stakes decisions, understanding the rationale behind their predictions is essential for fostering trust and ensuring responsible deployment. Explainable AI (XAI) has made substantial progress toward this objective, with approaches ranging from pixel-level saliency methods (Simonyan et al., 2013; Selvaraju et al., 2017) to global concept-based techniques (Ghorbani et al., 2019; Zhang et al., 2021). However, these paradigms typically operate in isolation: saliency methods provide spatially precise indications of *where* a model attends but offer limited semantic insight, whereas concept-based methods elucidate *what* semantic attributes (e.g., *"striped fur"*, *"blue beak"*) influence predictions but lack spatial specificity. Bridging these perspectives remains a central open challenge in visual interpretability. Furthermore, concept-based methods often rely on manual supervision (Kim et al., 2018) or clustering heuristics (Ghorbani et al., 2019), and few frameworks provide spatial grounding for the extracted concepts. This disconnect between semantic richness and spatial localization hinders the verification of whether identified concepts manifest in the input and limits the trustworthiness of resulting explanations. Critically, grounding explanations in coherent, human-aligned concepts may also mitigate the influence of spurious correlations, helping ensure that interpretability reflects meaningful visual factors rather than incidental background artifacts or dataset biases.

To this end, we introduce **Con**cept-based **Ex**planations (**ConEx**), a fully automatic, post-hoc framework that unifies faithfulness and interpretability without manual annotation or retraining. ConEx leverages label-free vision-language

---

[1]Tel Aviv University [2]The Open University. Correspondence to: Yehonatan Elisha <elisha@mail.tau.ac.il>.

*Proceedings of the 43rd International Conference on Machine Learning*, Seoul, South Korea. PMLR 306, 2026. Copyright 2026 by the author(s).

priors (Oikarinen et al., 2023) to extract class-discriminative textual attributes, grounds them into spatial masks via zero-shot segmentation (GroundedSAM (Ren et al., 2024)), and retains only those concepts satisfying principled thresholds for *occurrence rate* and *spatial coverage*. From these validated segments, it constructs robust Concept Activation Vectors (CAVs) using a centroid-difference formulation in the model's latent space, with architecture-specific embedding strategies for CNNs (layer-wise masking) and ViTs (unmasked patch aggregation). A multiplicative fusion of concept presence and model relevance yields pixel-accurate local explanations, while aggregated attributions provide class-level global explanations. Figure 1 illustrates our motivation. To evaluate the quality and reliability of learned concepts, we propose two complementary metrics: the Vector-Concept Match (VCM) and the Concept-Class Match (CCM). These metrics respectively measure the alignment between the learned CAVs and their visual meaning, and between the discovered concepts and the model's decision boundaries. We evaluate ConEx across three datasets, five model architectures, and four key dimensions: faithfulness (via perturbation tests and the FunnyBirds benchmark), concept quality (through concept-faithfulness and CAV validation metrics), segmentation accuracy, and human interpretability. The results show that ConEx consistently surpasses state-of-the-art saliency and concept-based explanation methods while remaining fully automated and scalable, effectively handling large-scale datasets such as ImageNet without requiring human supervision or model retraining. In summary, our contributions are:

(1) We introduce **ConEx**, a fully-automatic framework that bridges saliency-based visualization and concept-based reasoning, enabling both spatial and semantic interpretability.
(2) We introduce a CAV creation approach that eliminates the need for manual image curation for every concept by automatically discovering class-specific concepts and leveraging vision-language capabilities.
(3) We introduce architecture-specific embedding strategies that enhance concept purity and enables robust CAV construction for both CNNs and ViTs.
(4) We define two complementary metrics, VCM and CCM, to quantitatively assess concept reliability and alignment.
(5) We present comprehensive empirical validation demonstrating consistent gains in faithfulness, concept quality, spatial alignment, and human interpretability.

## 2. Related Work

**Post-Hoc Attribution Methods.** Interpretable AI has advanced rapidly in recent years, with significant developments across multiple modalities (Elisha et al., 2024; 2026b; Gurevitch et al., 2025; Barkan et al., 2023e;a; 2024c;b; 2026; Haddad et al., 2025; Arviv et al., 2026; Fong et al., 2019). Pixel attribution methods generate saliency maps to identify input regions influencing a model's prediction.

This family includes gradient-based approaches (Simonyan et al., 2013; Springenberg et al., 2015; Sundararajan et al., 2017; Srinivas & Fleuret, 2019; Barkan et al., 2025), which backpropagate class scores, activation-based methods such as Class Activation Maps (CAM) (Zhou et al., 2016) and its extensions (Wang et al., 2020; Ramaswamy et al., 2020), which leverage feature maps for spatial localization, and perturbation-based methods like RISE (Petsiuk et al., 2018), which measure sensitivity to input masking. Other works combine gradients with activations (Selvaraju et al., 2017; Chattopadhay et al., 2018; Barkan et al., 2021a;b), adopt game-theoretic formulations (Lundberg & Lee, 2017), employ path integration (Sundararajan et al., 2017; Barkan et al., 2023c;b;a;d), or apply relevance decomposition (Bach et al., 2015). Although these methods reveal *where* models focus, they fail to clarify *what* semantic concepts drive decisions. Feature visualization (Olah et al., 2017; 2018) attempts this but remains abstract and difficult to map to human-understandable concepts.

**Concept-Based Interpretability.** To provide semantic explanations, concept-based methods map model behavior to human-interpretable attributes. Network Dissection (Bau et al., 2017) aligns network units with labeled semantic concepts. Concept Activation Vectors (CAVs) (Kim et al., 2018) define directions in activation space corresponding to user-defined concepts, with TCAV (Kim et al., 2018) adding statistical validation. Subsequent work has aimed to automate this process. ACE (Ghorbani et al., 2019) uses unsupervised clustering to discover concepts, but its reliance on generic segments and full-image CAVs can conflate features. Other approaches explore generative manipulation (ICE (Zhang et al., 2021)) or model concept interactions (MCD (Vielhaben et al., 2023)). More recently, concept bottleneck models (Koh et al., 2020; Yuksekgonul et al., 2023) explicitly route predictions through a concept layer, but this typically requires extensive training-time annotations. A critical limitation of existing CAV methods is their construction: they often require manual curation (Kim et al., 2018) or use coarse image-level supervision (Ghorbani et al., 2019). Furthermore, classifier-based CAV training (e.g., linear SVMs) can be sensitive to sample selection and outliers (Martin & Weller, 2019). ConEx addresses these limitations via automatic, precisely-grounded concept discovery and a robust centroid-based CAV construction. A closely related line of work, Visual-TCAV (De Santis et al., 2024), bridges saliency and concept-based methods by pooling a difference-of-means CAV across spatial dimensions and using the resulting weights to produce concept localization maps. While effective, Visual-TCAV requires manual curation of concept example images, is restricted to user-specified concepts, and builds concept vectors using full images. ConEx overcomes these limitations by automatically discovering and validating class-discriminative concepts via vision-language priors and zero-shot segmentation, introducing architecture-specific embedding strategies

that yield purer CAVs, and employing multiplicative fusion with LRP for spatially precise, semantically grounded attribution.

**Architecture-Specific Interpretability.** Most interpretability research has primarily focused on CNNs, with considerably less attention given to ViTs. Although several transformer-specific attribution methods have been proposed (Chefer et al., 2021b; El-Nouby et al., 2021), concept-based interpretability frameworks remain largely centered on CNN architectures. The primary reason for this gap is the lack of spatial correspondence in ViTs, which makes generating localized explanations substantially more challenging and requires further dedicated research (Lee et al., 2024; Elisha et al., 2026a). Additionally, architectural differences, particularly in how spatial information is processed from irregular masks, necessitate specialized feature extraction strategies (Jain et al., 2022). In this work, we focus on addressing the second challenge, namely the need for architecture-specific feature extraction by introducing architecture-specific embedding strategies: layer-wise masking with neighborhood padding for CNNs and patch-based token selection and aggregation for ViTs. While our complete spatial explanation pipeline is currently designed for CNNs, the proposed CAV construction and validation framework is fully compatible with both architectures. This extension to ViTs is valuable in itself, as it provides a robust mechanism to quantitatively audit the global, semantic knowledge encoded in transformers. We demonstrate this compatibility empirically in Section 4.

## 3. Method

We introduce **ConEx (Concept-based Explanations)**, a framework that grounds image classifier predictions in semantically meaningful concepts. ConEx automatically discovers interpretable concepts, localizes them spatially, constructs their representations in latent space, and generates localized faithful explanations. Since ViTs lack inherent spatial correspondence, they require a dedicated formulation to enable localized concept-based explanations (Lee et al., 2024) - an open challenge that lies beyond the scope of this work. Nonetheless, we introduce and validate a dedicated CAV construction pipeline that produces high-quality global explanations for both ViTs and CNNs, outperforming existing approaches (Sec. 3.4), establishing a foundation for future ViT localization research. Figure 2 overviews the framework.

### 3.1. Problem Formulation

Let $f : \mathcal{X} \to \mathbb{R}^{|C|}$ be a pretrained image classifier mapping images $x \in \mathcal{X}$ to class logits over $C$ classes. Our goal is to explain $f$'s predictions through a set of human-interpretable concepts $\mathcal{K} = \{k_1, \ldots, k_m\}$, where each concept $k \in \mathcal{K}$ corresponds to a visual attribute (e.g., "floppy ears", "blue eyes"). For each concept, we seek: (1) a representation $\text{CAV}_k$ in the model's latent space, (2) a localization function $\text{M}_k(x)$ identifying where $k$ appears in $x$, and (3) an attribution score quantifying $k$'s contribution to predicting class $c$. The framework must operate without manual concept curation or additional training.

### 3.2. Automatic Concept Discovery and Validation

Given a dataset $\mathcal{D}$ with class set $C$, we generate a set of class-specific textual attributes $\text{TA}_c$ for each class $c \in C$ from its samples $\mathcal{D}_c$, following the concept generation procedure of (Oikarinen et al., 2023) ("Initial Concept Set Creation" in Fig. 2). The resulting attributes form an initial pool of linguistically interpretable candidate concepts. We then automatically validate and ground these concepts spatially. For an image $x$ with label $l$, the attribute set $\text{TA}_l$ is passed to GroundedSAM (Ren et al., 2024), which produces segmentation masks $\text{SM}_k(x)$ for visually detected concepts $k$ (and $\emptyset$ otherwise). A concept is retained only if it satisfies two criteria: **(1) Occurrence Rate**, defined as $\frac{|\{x \in \mathcal{D}_c : \text{SM}_k(x) \neq \emptyset\}|}{|\mathcal{D}_c|}$, and **(2) Spatial Coverage**, measured as the average IoU between $\bigcup_k \text{SM}_k$ and the class-specific segmentation masks. These filters yield a compact set of frequently occurring, spatially grounded concepts with strong human interpretability. Additional details are provided in the Appendix.

### 3.3. Concept Activation Vectors

For each validated concept $k$, we construct a Concept Activation Vector (CAV) from a set of example images representing the concepts. This CAV captures its direction in the model's latent space. Following (Kim et al., 2018), we collect $N$ positive samples from masks where $k$ is present $(\text{SM}_k)$ and $N$ negative samples from masks of other concepts $(\text{SM}_j, j \neq k)$. Let $\mathcal{E}_k^+$ and $\mathcal{E}_k^-$ denote the sets of embeddings (see Sec. 3.4) for positive and negative samples, respectively. The CAV is computed as the difference of mean embeddings:

$$\text{CAV}_k = \mu_k^+ - \mu_k^-, \quad \text{where} \quad \mu_k^\pm = \frac{1}{|\mathcal{E}_k^\pm|} \sum_{e \in \mathcal{E}_k^\pm} e. \quad (1)$$

This simple centroid difference is more robust to perturbations than classifier-based approaches (e.g., SVM) as shown in Martin & Weller (2019). Specifically, the method computes the arithmetic mean of the activations for the positive samples $(\mathcal{E}_k^+)$ and the negative samples $(\mathcal{E}_k^-)$, and then directly computes the CAV as the difference between these centroids. The resulting $\text{CAV}_k$ represents meaningful direction in the activations of a layer in a neural network (Martin & Weller, 2019).

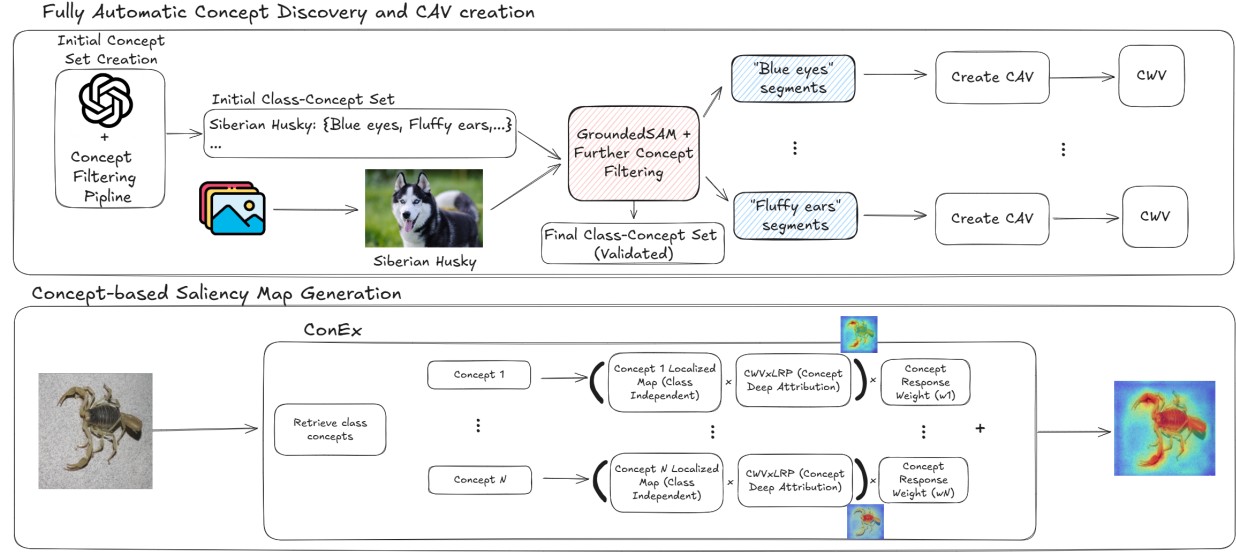

*Figure 2.* The ConEx Framework: (top) Automatic discovery of class-specific concepts and CAV construction. (bottom) Generation of concept-based saliency maps.

## 3.4. Architecture-Specific Embedding Strategies

We adopt embedding strategies tailored to CNNs and ViTs that preserve semantic information while avoiding masking artifacts. A more detailed description and ablation analyses are provided in the Appendix.

**CNNs.** Embedding irregularly shaped concept segments in CNNs is non-trivial: naïve strategies such as zero-filling or color-filling introduce boundary artifacts that contaminate the resulting representations (Ghorbani et al., 2019). We instead adopt *layer-wise masking* (Balasubramanian & Feizi, 2023), propagating both the image and its binary segment mask through the network and retaining, at each layer, only activations derived from unmasked pixels. To prevent boundary erosion without leaking context from masked regions, we apply *neighborhood padding* at the first convolutional layer only: boundary pixels are filled with the local mean of their unmasked neighbors, iterated to match the kernel size. Confining padding to a single early layer grants limited contextual access while preserving fine-grained segment boundaries - crucial for precisely segmented attributes such as beaks or eyes.

**ViTs.** For ViTs, images are patch embeddings interacting via global self-attention. We extract embeddings from an intermediate layer, retaining only patch embeddings overlapping the segment, then average them. Intermediate layers balance spatial detail and semantic abstraction (Raghu et al., 2021; Caron et al., 2021), yielding stable representations. This mirrors CNN's localized extraction, ensuring cross-architecture consistency.

## 3.5. Concept Aware Attribution

### 3.5.1. CONCEPT LOCALIZATION VIA CHANNEL-WEIGHTED VECTORS

To localize concept $k$ in an arbitrary image $x$, we compute its *channel-weighted vector* (CWV) by globally average pooling $CAV_k$ across spatial dimensions, resulting with $CWV(k) = \{CWV_1(k), \ldots, CWV_h(k)\}$, where $h$ indexes channels. Each scalar $CWV_h(k)$ approximates the correlation between feature map $h$ and concept $k$, independent of spatial location. We generate a raw semantic map by computing a weighted sum between $CWV(k)$ and the latent representation $Latent(x)$ of image $x$ over the channels:

$$M'_k(x) = \text{ReLU}\left( \sum_h CWV_h(k) \cdot Latent_h(x) \right). \quad (2)$$

ReLU filters regions that are uncorrelated with the concept. To enable cross-concept and cross-image comparisons, we normalize $M'_k(x)$ by the maximum activation of the positive centroid:

$$M_{(i,j)k}(x) = \min\left( 1, \frac{M'_{(i,j)k}(x)}{nv_k + \epsilon} \right), \quad (3)$$

where

$$nv_k = \max\left( \text{ReLU}\left( \sum_h CWV_h(k) \cdot \mu_k^+ \right) \right) \quad (4)$$

where $i, j$ refer to spatial dimensions of the feature maps and $\epsilon$ prevents division by zero. The normalized map

$M_{(i,j)k}(x) \in [0, 1]$ highlights regions where the network recognizes concept $k$, with intensity indicating activation strength. This provides class-agnostic, interpretable visualization of concept presence. CWV is mechanistically related to the pooled-CAV of Visual-TCAV; however, two design choices yield materially stronger representations. First, our CAVs are constructed from automatically discovered, precisely segmented concept regions embedded via architecture-specific strategies (Sec. 3.4), rather than from manually curated full images - eliminating background confounds that dilute concept directions. Second, normalization is anchored to the positive concept centroid (Eq. 3), enabling cross-concept and cross-image comparability.

### 3.5.2. CONCEPT-AWARE ATTRIBUTION VIA MULTIPLICATIVE FUSION

To quantify how concepts influence class decisions, we integrate semantic maps with Layer-wise Relevance Propagation (LRP) (Bach et al., 2015). For class $c$, let $S_h^c(x)$ denote the LRP attribution. We compute the *concept importance map* as:

$$\text{CIM}_k^c(x) = M_k(x) \cdot \left( \sum_h \text{ReLU}(\text{CWV}_h(k)) \cdot S_h^c(x) \right).$$
(5)

Multiplicative fusion amplifies attributions only where concept presence ($M_k$) and model relevance ($S^c$) coincide, suppressing irrelevant regions for spatially precise, semantically grounded explanations. Notably, extending this approach to multi-label classification is straightforward, as each concept corresponds to a class-discriminative attribute rather than to a mutually exclusive category.

### 3.5.3. CLASS-SPECIFIC SEMANTIC ATTRIBUTION

To measure the contribution of concept $k$ to predicting class $c$, we first define the predicted probability for class $c$ as $p_c(x) = [\text{softmax}(f(x))]_c$, where $f(x)$ is the logit vector from the model. We then perform an intervention by masking the concept and observing the change in this probability:

$$w_k^c(x) = \frac{p_c(x) - p_c(x \setminus k)}{p_c(x) + \epsilon},$$
(6)

where $p_c(x \setminus k)$ is the probability when regions corresponding to concept $k$ (identified by $M_k(x) > \tau$, where $\tau$ is the mean value) are zeroed. This normalized drop quantifies the concept's influence on the model's prediction. The final saliency map aggregates all concepts associated with class $c$:

$$\text{FinalMap}_c(x) = \sum_{k \in \text{TA}_c} w_k^c(x) \cdot \text{CIM}_k^c(x).$$
(7)

This provides a concept-decomposed explanation of the classifier's decision, revealing which concepts drove the prediction and where they were detected.

### 3.5.4. GLOBAL SEMANTIC ATTRIBUTION

Beyond per-image explanations, ConEx produces global insights by aggregating semantic attributions across the dataset. For concept $k$ and class $c$, we compute the global importance as:

$$G_k^c = \frac{1}{|\mathcal{D}_c|} \sum_{x \in \mathcal{D}_c} \sum_{i,j} \text{CIM}_{k(i,j)}^c(x),$$
(8)

where the inner sum is over spatial locations. High $G_k^c$ indicates that concept $k$ consistently contributes to predictions of class $c$ across many examples, enabling interpretable class-level characterizations (e.g., identifying "blue eyes" as globally important for "siberian husky"). Global attributions complement local maps, providing both instance-specific and population-level understanding.

### 3.6. CAV Validation Metrics

We introduce two metrics to validate CAV quality: **Vector-Concept-Match (VCM)** and **Concept-Class-Match (CCM)**. For evaluating ConEx with these metrics, we use $\text{CWV}(k)$ in place of the CAV, as it provides an accurate representation of the concept's direction within the latent feature space.

**VCM.** quantifies whether $\text{CAV}_k$ aligns with its intended concept. For a test set $T_k$ of segments containing concept $k$, we compute:

$$S_{\text{VCM}}(k) = \frac{1}{|T_k|} \sum_{x \in T_k} \frac{\text{CAV}_k \cdot \text{Latent}(x)}{\|\text{CAV}_k\| \|\text{Latent}(x)\|}.$$
(9)

Higher VCM indicates strong CAV-concept alignment.

**CCM.** evaluates whether $\text{CAV}_k$ predicts its associated class $c$. For a set of concept-class pairs $Q$, we compute:

$$S_{\text{CCM}} = \frac{1}{|Q|} \sum_{(k,c) \in Q} \frac{\text{CAV}_k \cdot \nabla f_c(x)}{\|\text{CAV}_k\| \|\nabla f_c(x)\|},$$
(10)

where $\nabla f_c(x)$ is the gradient of the class logit with respect to the latent representation. Higher CCM confirms that the concept not only exists in the feature space but also positively influences the class decision. Together, VCM and CCM provide complementary validation of semantic fidelity and predictive relevance.

## 4. Experiments

### 4.1. Experimental Setup

**Datasets, Models, and Baselines.** We conduct experiments on ImageNet **(IN)** (Deng et al., 2009), CUB-200-2011 **(CUB)** (Wah et al., 2011), Stanford Dogs **(SD)** (Khosla et al., 2011), and Imagenet-Segmentation **(IN-S)**. We eval-

uate five pretrained models: ResNet50 (**RN**) (He et al., 2016), DenseNet121 (**DN**) (Huang et al., 2017), ConvNeXt-Base (**CN**) (Li et al., 2022), ViT-Base (**ViT-B**), and ViT-Small (**ViT-S**) (Dosovitskiy et al., 2021). All models were sourced from the `timm` library and utilize their standard corresponding dataset pretrained weights. We compare ConEx against state-of-the-art saliency methods: Grad-CAM (**GC**) (Selvaraju et al., 2017), Integrated Gradients (**IG**) (Sundararajan et al., 2017), Score-CAM (**SC**) (Wang et al., 2020), FullGrad (**FG**) (Srinivas & Fleuret, 2019), **RISE** (Petsiuk et al., 2018), AblationCAM (**AC**) (Ramaswamy et al., 2020), **SHAP** (Lundberg & Lee, 2017), and Integrated Iterated Attributions (**IIA**) (Barkan et al., 2023b). In addition, we evaluate ConEx against prominent concept-based frameworks: Automated Concept-based Explanations (**ACE**) (Ghorbani et al., 2019), Invertible Concept-based Explanations (**ICE**) (Zhang et al., 2021), and Multi-Concept Decompositions (**MCD**) (Vielhaben et al., 2023). All baselines use official implementations with author-recommended hyperparameters to ensure a fair comparison.

**Implementation Details. (1) Concept discovery and grounding.** We used GroundedSAM (Ren et al., 2024) with occurrence rate $\geq 15\%$ and spatial coverage $\geq 20\%$, yielding 348 concepts (CUB), 189 (SD), and 3,602 (IN). Initial concepts from Oikarinen et al. (2023) were created using GPT-4o. **(2) CAV construction.** $N = 100$ positive/negative segments, computed as centroid differences (Eq. 1). For CNNs, embeddings extracted from final convolutional layer with 7×7 neighborhood padding. For ViTs, we derive unmasked patch-token embeddings by averaging patches from layer 6, balancing spatial fidelity and semantic abstraction (Raghu et al., 2021; Caron et al., 2021) (see Appendix for layer ablation). **(3) Attribution integration.** LRP (Bach et al., 2015) with $\epsilon$-rule ($\epsilon = 0.01$), multiplicative fusion for concept importance maps, threshold $\tau$ set to mean activation value.

All experiments are conducted on NVIDIA A100 GPUs using PyTorch. The Appendix provides additional implementation details.

### 4.2. Evaluation Protocols

#### 4.2.1. FAITHFULNESS

We employ two complementary protocols: **(1) Perturbation Metrics:** Standard Insertion (**INS**) and Deletion (**DEL**) (Petsiuk et al., 2018; Barkan et al., 2024a; Baklanov et al., 2025). **(2) FunnyBirds Benchmark:** To mitigate the domain-shift problem inherent in pixel-perturbation methods, we evaluate FunnyBirds (Hesse et al., 2023) (500 images, 50 classes), using the provided RN pretrained model[1] and predefined parts (beak, wings, feet, eyes, and tail) as concepts for

[1] https://github.com/visinf/funnybirds/tree/main/

ConEx. This protocol uses controlled, part-based interventions to provide a ground-truth importance score for object parts. For these experiments, we evaluate top 5 performing methods from Tab 1. We report Completeness (**CMP**), Correctness (**CRC**), and Contrastivity (**CNT**) scores, which assess if explanations cover all important parts, only important parts, and distinguish between classes, respectively. Specific details are provided in the Appendix. While a direct quantitative comparison with Visual-TCAV would be natural, it is not straightforward. Visual-TCAV generates saliency maps for a single user-specified concept at a time, whereas faithfulness metrics such as INS/DEL evaluate the completeness of an explanation with respect to the model's prediction. As no single concept is expected to fully explain the prediction, directly applying these metrics would systematically disadvantage Visual-TCAV. To enable a fairer comparison, we additionally evaluate Visual-TCAV using ConEx's aggregation mechanism in Sec. 4.4.

#### 4.2.2. CONCEPT QUALITY

For every model and method pair, a set of CAVs were created. The VCM and CCM evaluations are conducted using a 20% held-out test set of concept segments from IN (not used during CAV construction). We report evaluation results on the RN, DN, CN, ViT-B, and ViT-S models. Unless stated otherwise, we follow the original implementations of ACE, ICE, and MCD.

**Concept Insertion and Deletion (CINS/CDEL).** Following ACE (Ghorbani et al., 2019), we use 100 random IN classes and 1,000 images, measuring AUC as concepts are progressively revealed (CINS) or masked (CDEL). For each validation image, concept masks are generated using SAM, latent activations and local importance scores are computed following Vielhaben et al. (2023). When local importance scores are unavailable for a method, we adopt the protocol in Vielhaben et al. (2023) and use TCAV scores. For CNN-based evaluations of ACE, ICE, and MCD, we adhere to their original implementation details, noting that ACE relies on TCAV scores rather than local concept importance scores. ViT-based evaluations also employ TCAV scores for concept importance to account for the lack of spatial correspondence in ViTs. Final results are averaged over the 1,000 randomly sampled images.

**VCM and CCM.** Both metrics are defined in Sec. 3.6. For evaluating ConEx, we use the CWV as the CAV, as it reliably captures the direction of each concept within the latent representation space. For VCM, we reserve 20% of concept segments as a test set and report average similarity. We collected a set of 200 concepts (25 segments were sampled from each) with *occurrence rates* of at least 25%, and from varying classes. Segments were encoded using the same mechanism described in Sec. 3.4. For CCM, we select 50 distinct classes, extract 5 representative concepts per class, and sample 20 segments per concept, yielding a total of

5,000 segment samples. Hence, we report average cosine similarity between CAVs and class gradients over 100 randomly sampled images per class.

### 4.2.3. SEGMENTATION ALIGNMENT.

We evaluate the RN, DN, and CN models on the IN-S dataset, using the top five methods from Tab. 1, following (Barkan et al., 2023b; Chefer et al., 2021b), to assess how well each explanation's spatial distribution aligns with human-annotated object regions. Further details and results are provided in the Appendix.

### 4.2.4. HUMAN EVALUATION PROTOCOLS

We asses Understandability (Vielhaben et al., 2023) and class-level concept sets quality. Complete methodological details and concept quality results are provided in the Appendix.

**Understandability.** Following (Zhang et al., 2021), and inspired by task prediction protocols (Hoffman et al., 2023), participants view a test image with one concept highlighted and select the best match from five candidate concept explanations from same class. We follow the standard protocol, evaluating the following metrics: Selection Accuracy (**SA**), Percentage of Recognizable Concepts (**PPR**), Inner-Concept Description Similarity (**INNS**), and Intra-Concept Description Similarity (**INTS**). A detailed description of all metrics is provided in the Appendix. The study followed a within-subject design. We evaluated 15 examples drawn from three methods (ACE, MCD, and ConEx) across five randomly selected IN classes. Each participant viewed three examples per class, with method order randomized. For each method, explanations were generated by retaining the ten most influential concepts per class, each represented by its ten most prototypical samples. From these, five candidate concepts were randomly selected for each test image. Ten random samples were created per class and method, yielding a total of 300 distinct samples. Each participant received a unique set and order of samples and completed a brief tutorial. Fifty-eight participants completed the survey, which lasted approximately 30 minutes.

### 4.3. Results and Analysis

In all tables, the best results are in **bold**, and the second-best are underlined.

### 4.3.1. FAITHFULNESS ANALYSIS

As shown in Table 1, ConEx achieves state-of-the-art performance across all perturbation-based faithfulness metrics (INS/DEL), datasets, and architectures. Results for the SD dataset are provided in the Appendix. This is particularly noteworthy as ConEx operates at the *concept level*, yet still surpasses pixel-based methods like GC, SC, and IIA. The high INS and low DEL scores confirm that ConEx expla-

nations capture the model's true decision-making regions and are effective at excluding spurious correlations, which is a limitation commonly observed in pixel-based saliency methods (Bertrand et al., 2022). This synergy of localization (CWVs) and relevance (Eq. 5) allows ConEx to be both semantically meaningful and faithfully aligned with model decisions. This robustness is further confirmed in Table 2, where ConEx also achieves state-of-the-art performance on the part-based FunnyBirds benchmark. The distinct evaluation protocol employed in FunnyBirds offers a complementary perspective to the faithfulness evaluation, further reinforcing our confidence in the robustness of ConEx by showing that it achieves high performance also using a pre-defined set of concepts. This strong performance underscores that explaining a model's decision through localized semantic attributions can still capture the most influential regions, thereby promoting both interpretability and faithfulness.

### 4.3.2. CONCEPT QUALITY AND CAV VALIDATION

Table 3 show ConEx's superior concept quality. We achieve the highest scores on concept-level faithfulness (CINS and CDEL) and on our proposed validation metrics (VCM and CCM) across all architectures, including ViTs. Unlike ACE or ICE, our CAVs are built from precisely grounded segments using a specified embedding strategy and a robust centroid-difference formulation (Martin & Weller, 2019). The high VCM scores suggest that our CAVs provide better semantic purity, while the high CCM scores confirm they are also *predictively relevant*, directly influencing the model's class decision. These findings support our core hypothesis that encoding concept segments accurately within the image leads to meaningful concept vectors. Finally, we observe that ViT-based models exhibit comparatively lower scores than CNN counterparts. We attribute this to the inherent challenges of operating on patch-level representations in ViTs, as previously discussed on Sec. 3.

### 4.3.3. HUMAN INTERPRETABILITY

ConEx demonstrates superior performance in the understandability evaluation (Table 4). It achieves the highest scores for SA, PPR, and INNS, and lowest INTS. This demonstrates that our explanations are not only faithful to the model but also more intuitive, consistent, and semantically distinct to human users than prior state-of-the-art methods. This connection between human interpretability and model relevance supports the growing perspective that faithfulness in explanations can also enhance user comprehensibility (Doshi-Velez & Kim, 2017; Lipton, 2018).

### 4.4. Qualitative Comparison and Ablation Studies

**Qualitative Analysis.** Figure 3 provides a qualitative comparison on IN. ConEx explanation maps are clearly more localized and semantically coherent than pixel-based

*Table 1.* Faithfulness Perturbation Evaluation: Comparison of Insertion (INS ↑) and Deletion (DEL ↓) AUC scores across datasets and models.

| Dataset | Model | Metric | GC | SC | SHAP | AC | RISE | FG | IG | IIA | ConEx |
|---------|-------|--------|-----|-----|------|-----|------|-----|-----|------|-------|
| CUB | RN | INS | 55.73 | 57.64 | 46.92 | 56.82 | 54.67 | 50.48 | 48.97 | 58.13 | **59.47** |
| | | DEL | 12.63 | 11.82 | 15.51 | 12.14 | 13.28 | 13.66 | 11.42 | 9.21 | **8.19** |
| | DN | INS | 56.18 | 58.09 | 47.31 | 57.22 | 55.04 | 50.93 | 49.58 | 58.81 | **59.62** |
| | | DEL | 12.24 | 11.53 | 15.35 | 11.87 | 12.92 | 13.45 | 11.19 | 8.96 | **8.03** |
| | CN | INS | 54.96 | 57.51 | 46.63 | 56.08 | 54.24 | 50.21 | 48.73 | 58.48 | **59.12** |
| | | DEL | 12.71 | 11.79 | 15.64 | 12.12 | 13.17 | 13.71 | 11.33 | 9.68 | **8.25** |
| IN | RN | INS | 54.81 | 56.92 | 47.03 | 56.31 | 53.96 | 50.72 | 49.12 | 58.03 | **59.11** |
| | | DEL | 12.74 | 11.97 | 15.48 | 12.41 | 13.26 | 13.67 | 11.54 | 9.53 | **8.49** |
| | DN | INS | 55.12 | 57.34 | 47.29 | 56.58 | 54.21 | 50.93 | 49.36 | 58.76 | **59.32** |
| | | DEL | 12.51 | 11.84 | 15.27 | 12.28 | 13.06 | 13.49 | 11.41 | 9.62 | **8.33** |
| | CN | INS | 54.23 | 56.47 | 46.81 | 55.86 | 53.68 | 50.54 | 48.92 | 58.12 | **58.77** |
| | | DEL | 12.96 | 12.19 | 15.55 | 12.54 | 13.42 | 13.81 | 11.67 | 9.35 | **8.44** |

*Figure 3.* Qualitative Results: Explanation maps produced using RN w.r.t. the classes (top to bottom): 'wing', 'hornbill', 'motor scooter, scooter', 'convertible'.

*Table 2.* FunnyBirds Benchmark: Comparing Completeness (CMP), Correctness (CRC), and Contrastivity (CNT) scores using the RN model.

| Metric | RISE | GC | AC | SC | IIA | ConEx |
|--------|------|-----|------|------|------|-------|
| CMP ↑ | 0.68 | 0.70 | 0.76 | 0.72 | 0.74 | **0.78** |
| CRC ↑ | 0.54 | 0.56 | 0.61 | 0.57 | 0.59 | **0.64** |
| CNT ↑ | 0.63 | 0.68 | 0.83 | 0.85 | 0.88 | **0.92** |

methods (GC, FG), which often highlight diffuse regions. Our maps align cleanly with object parts, allowing a user to understand *which* attribute (e.g., "yellow beak" for "hornbill") drove the prediction, not just *where* the model looked.

**Ablation Studies.** We provide ablation studies validating the key design choices of ConEx. Specifically, we show

that: **(1)** Our centroid-based CAVs built from accurately embedded segments are more faithful than CAVs created using ACE, ICE, or MCD methodologies. **(2)** Our multiplicative fusion (Eq. 5) with LRP outperforms using other attribution maps (GC, SHAP, IIA) and provides a significant improvement over the base LRP map alone. Moreover, we observe that applying ConEx with every saliency method (GC, SHAP, IIA) improves performance on faithfulness metrics. **(3)** CAVs built from concept-specific segments are decisively superior to those built from full images containing the concept. **(4)** We analyze the sensitivity to $N$, the number of samples used for CAV creation. Due to space constraints, only results (1) and (2) are included in the main paper, while the remaining analyses are provided in the Appendix.

**(1) CAV Construction Strategy Comparison.** We assess

*Table 3.* Concept Quality and Faithfulness Evaluation: results for Concept Faithfulness (CINS, CDEL) and our proposed CAV validation metrics (VCM, CCM) on IN dataset across all models.

| Metric | ViT-B | | | | ViT-S | | | | RN | | | | DN | | | | CN | | | |
|---|---|---|---|---|---|---|---|---|---|---|---|---|---|---|---|---|---|---|---|---|
| | ACE | ICE | MCD | ConEx | ACE | ICE | MCD | ConEx | ACE | ICE | MCD | ConEx | ACE | ICE | MCD | ConEx | ACE | ICE | MCD | ConEx |
| CINS ↑ | 30.97 | 30.18 | 33.23 | **35.33** | 30.08 | 29.31 | 32.27 | **34.30** | 44.89 | 43.74 | 48.16 | **51.20** | 44.63 | 43.58 | 47.92 | **51.37** | 44.75 | 43.62 | 48.03 | **51.44** |
| CDEL ↓ | 13.15 | 13.70 | 11.93 | **10.35** | 13.31 | 13.87 | 12.34 | **10.51** | 10.82 | 11.28 | 9.56 | **8.74** | 10.93 | 11.41 | 9.48 | **8.66** | 10.87 | 11.36 | 9.59 | **8.72** |
| VCM ↑ | 0.38 | 0.34 | 0.42 | **0.53** | 0.37 | 0.33 | 0.41 | **0.51** | 0.53 | 0.49 | 0.56 | **0.68** | 0.52 | 0.48 | 0.57 | **0.69** | 0.52 | 0.48 | 0.56 | **0.67** |
| CCM ↑ | 12.31 | 10.18 | 11.49 | **18.43** | 11.76 | 10.05 | 11.63 | **17.92** | 23.04 | 21.38 | 20.22 | **28.01** | 22.48 | 20.91 | 20.17 | **27.63** | 22.27 | 21.01 | 21.35 | **27.41** |

*Table 4.* Human Evaluation: understandability tests on IN dataset using RN model over SA (%), PRC (%), INNS, and INTS.

| Method | SA ↑ | PRC ↑ | INNS ↑ | INTS ↓ |
|---|---|---|---|---|
| ACE | 58.47 | 49.51 | 0.41 | 0.34 |
| MCD | 34.32 | 62.28 | 0.47 | 0.39 |
| ConEx | **72.13** | **70.48** | **0.52** | **0.28** |
| ANOVA test p-values | < 0.001 | < 0.001 | < 0.001 | < 0.001 |
| **T-test p-values** | | | | |
| ConEx vs ACE | < 0.001 | < 0.001 | < 0.001 | 0.0149 |
| ConEx vs MCD | < 0.001 | 0.001 | 0.006 | < 0.001 |
| ACE vs MCD | 0.028 | 0.3549 | 0.4143 | < 0.001 |

*Table 5.* Ablation on CAV creation mechanism: faithfulness tests comparing different methods using the IN dataset and RN model.

| Method | Con-ACE | Con-ICE | Con-Vis | Con-MCD | ConEx |
|---|---|---|---|---|---|
| INS ↑ | 57.45 | 57.92 | 56.98 | 58.16 | **59.21** |
| DEL ↓ | 9.58 | 9.83 | 10.61 | 10.22 | **8.34** |

*Table 6.* Ablation on explanation method fusion: faithfulness tests comparing different explanation methods using the IN dataset and RN model.

| Method | INS ↑ | DEL ↓ |
|---|---|---|
| GC | 54.92 | 12.68 |
| SHAP | 47.54 | 15.49 |
| IIA | 58.39 | 9.48 |
| Con-GC | 58.68 | 8.92 |
| Con-SHAP | 56.99 | 9.84 |
| Con-IIA | 58.96 | 8.65 |
| ConEx | **59.21** | **8.34** |

the effectiveness of our CAV construction strategy by comparing it to the strategies used in ACE, ICE, Visual-TCAV, and MCD. To enable a fair comparison, we implemented four additional ConEx variants (**Con-ACE, Con-ICE, Con-Vis, Con-MCD**), each replacing our CAV construction mechanism with that of the corresponding baseline method. For each variant, CAVs were generated using the exact procedures described in the original implementations of ACE, ICE, Visual-TCAV, and MCD. Table 5 reports the results (with the name of the methods indicating their ConEx adjusted version). The findings demonstrate a clear advantage for our CAV construction approach, which we attribute to its concept-centered design, leveraging concept segments, and to our embedding strategy, which preserves the most informative features of each segment.

**(2) Explanation Methods for Attribution Fusion.** We evaluate the benefit of using LRP within the fusion strategy described in 3.5.2, comparing it against alternative explanation methods such as GC, SHAP, and IIA. To conduct this analysis, we implemented three additional ConEx variants (**Con-GC**, **Con-SHAP**, **Con-IIA**), each replacing the LRP-based multiplicative fusion step with the corresponding baseline method. Table 6 presents the results (method names reflect their adjusted ConEx variants). The findings show a clear advantage for LRP in this fusion stage, likely because LRP propagates relevance through multiple layers, offering a more comprehensive view of the model's internal representations. Furthermore, across all methods, integrating them into the ConEx framework yields substantial performance gains over their standalone versions, highlighting the overall effectiveness of our approach.

## 5. Conclusion

This work presented **ConEx**, a fully automatic and concept-driven framework for visual explanations that unifies semantic interpretability and model faithfulness. By integrating label-free vision-language priors with zero-shot segmentation, ConEx discovers, grounds, and validates class-discriminative concepts without human supervision or retraining. Through extensive experiments across multiple datasets and architectures, we demonstrated that ConEx delivers precise spatial grounding, high semantic fidelity, and strong quantitative performance, consistently outperforming existing saliency and concept-based methods. These results highlight that concept-based interpretability, when properly grounded in the model's representational space, can yield explanations that are simultaneously human-meaningful, faithful, and robust to spurious correlations. Limitations and future work are discussed in the Appendix.

## Acknowledgment

This work was supported by the Ministry of Innovation, Science & Technology, Israel.

## Impact Statement

This paper presents work whose goal is to advance the field of machine learning by improving the interpretability of computer vision models. By bridging saliency visualizations with concept-based reasoning to provide spatially grounded explanations, ConEx has the potential to yield significant positive societal impacts. Specifically, it can assist practitioners in auditing high-stakes AI decision-making systems, fostering user trust, and identifying harmful dataset biases or spurious correlations. However, because our automated concept discovery pipeline relies on Large Language Models to generate initial class-discriminative textual attributes, there is a potential risk that societal biases encoded within the vision-language priors could be inadvertently transferred into the generated explanations. We encourage practitioners to remain mindful of these inherited biases when deploying this framework in fairness-critical applications to ensure that the interpretability tools themselves do not perpetuate unintended harms.

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

# A. Appendix Overview

This appendix presents supplementary materials, implementation specifics, and extended analyses to support the findings discussed in the main paper. The content is organized as follows:

- **Section B** outlines the implementation details and technical specifications.

- **Section C** elaborates on the concept set validation process and details the comparative analysis that informed our final configuration choices.

- **Section D** explores optional enhancements for ConEx, such as the construction of a rigorous negative sample set to refine the CAV creation process.

- **Section E** outlines in detail the challenges of producing localization maps for ViTs.

- **Section F** provides additional faithfulness results for the SD dataset.

- **Section G** provides additional experimental details and results for the human evaluation of class-specific concept sets.

- **Section H** presents an additional ablation analysis.

- **Section I** provides further experimental settings and results regarding segmentation tests and the FunnyBirds benchmark (Hesse et al., 2023).

- **Section J** discusses current limitations and suggests potential avenues for future research.

# B. Implementation Details

**Concept Grounding.** Concepts are automatically discovered and grounded using GroundedSAM (Ren et al., 2024) with a box threshold of 0.3 and a text threshold of 0.25.

**CAV construction.** For each validated concept, we construct Concept Activation Vectors by sampling $N = 100$ positive segments (regions where the concept is present) and $N = 100$ negative segments (regions containing other concepts from the randomly selected classes). CAVs are computed as the difference of mean embeddings (Eq. 1).

**Embedding extraction.** For CNNs, we extract embeddings from the final convolutional layer (pre-global pooling) using layer-wise masking (Balasubramanian & Feizi, 2023). To preserve semantic information while avoiding boundary artifacts, we apply 7×7 neighborhood padding at the first convolutional layer.

**Concept map generation.** Channel-weighted vectors (CWVs) are computed by global average pooling over CAV spatial dimensions. Concept maps are normalized using in Eq. 3 with $\epsilon = 10^{-8}$. Finally, during the map aggregation step (Eq. 7), we apply a clipping procedure to prevent overflow in regions where multiple concepts overlap.

**Attribution integration.** We use Layer-wise Relevance Propagation (LRP) (Bach et al., 2015) with $\epsilon$-rule ($\epsilon = 0.01$). Concept importance maps are computed via multiplicative fusion (Barkan et al., 2023b). The masking threshold $\tau$ in Eq. 6 is set to the mean value.

**Masking Strategies.** To ensure robustness of our evaluation, we examined alternative masking strategies for both concept intervention (Eq. 6) and perturbation-based metrics. Following Sturmfels et al. (2020), we evaluated four masking baselines: (1) zero baseline (black image), (2) Gaussian noise sampled from $\mathcal{N}(\mu, \sigma^2)$ matching IN statistics, (3) uniform noise sampled from $\mathcal{U}(0, 1)$, (4) Gaussian blur, and (5) training data baseline using randomly sampled patches from the dataset. Across all baselines, the relative performance trends remained consistent. ConEx maintained superior faithfulness scores compared to all baselines, with performance differences between methods varying by less than 2% across masking strategies.

**ViT Embeddings.** Unlike CNNs, Vision Transformers (ViTs) operate on patch embeddings that interact through global self-attention. This structure enables strong global reasoning but makes the model highly sensitive to alterations in token composition. Naïvely discarding or zeroing out tokens corresponding to masked regions disrupts the pretrained attention structure and leads to unstable or semantically inconsistent embeddings. To address this, we adopt an *intermediate feature extraction* strategy that isolates the segment's visual content while preserving the model's representational integrity.

Specifically, we first compute the full set of patch embeddings for the unaltered image using a pretrained ViT encoder. The segmentation mask is then projected onto the patch grid, and only the embeddings of patches overlapping the segment are retained. Rather than relying on the global [CLS] token, which aggregates information from the entire image and is thus influenced by surrounding context, we average the retained patch embeddings from an intermediate layer, typically chosen from the middle of the encoder stack. Intermediate representations have been shown to balance fine-grained spatial detail with higher-level semantics (Raghu et al., 2021; Caron et al., 2021), resulting in stable and semantically meaningful embeddings.

Notably, our experiments confirm that this masked averaging strategy yields superior CAV quality compared to the standard [CLS] token approach.

Conceptually, this approach parallels the localized feature extraction strategy used for CNNs: both architectures derive embeddings directly from spatially corresponding activations rather than from global aggregation tokens. By leveraging intermediate ViT features restricted to the segment, we ensure that each concept embedding reflects the internal visual evidence of the segment while remaining consistent with the model's learned feature hierarchy.

## C. Concept Set Validation

Following the initial concept generation using the label-free approach from (Oikarinen et al., 2023), we refined the results to ensure a high-quality concept set. Specifically, we applied filtering thresholds requiring an occurrence rate of $\geq 15\%$ and a spatial coverage of $\geq 20\%$. Spatial coverage was measured w.r.t. the object segment of the maximum predicted class.

**A detailed description of the validation process.** The following is a detailed version of Sec. 3.2. For a dataset $\mathcal{D}$ with classes $C$, we extract discriminative textual attributes $\text{TA}_c$ for each class $c \in C$ following the concept set creation procedure in (Oikarinen et al., 2023) (denoted as "Initial Concept Set Creation" in Fig. 2). This yields candidate concepts that are class-discriminative and linguistically interpretable. To ensure concept reliability, we perform a further automated concept validation during the grounding phase. Given an image $x$ with label $l$ (or predicted label if unavailable), we pass its corresponding attribute set $\text{TA}_l$ to GroundedSAM (Ren et al., 2024), a zero-shot grounding model combining GroundingDINO (Liu et al., 2024) with SAM (Kirillov et al., 2023). For each concept $k \in \text{TA}_l$, GroundedSAM returns corresponding segmentation masks when the concept is visually present, and no mask otherwise. Thus, for every image $x$ and a concept $k$ this phase results with a set $\text{SM}_k(x)$ which is empty in case $k$ is not present in $x$. We validate concepts via two criteria: (1) **Occurrence Rate**: the fraction of class images containing $k$, computed as $|\{x \in \mathcal{D}_c : \text{SM}_k(x) \neq \emptyset\}|/|\mathcal{D}_c|$, and (2) **Spatial Coverage**: the average IoU between $\bigcup_k \text{SM}_k$ and class-specific segmentation masks, i.e., the extent to which the concepts visually cover the segment associated with their class. The object segments are created using GroundedSAM with the class label as the prompt. Concepts failing either threshold are discarded. This yields a filtered set of spatially grounded, frequently occurring concepts per class with high human judgment alignment.

**Example Discovered Concepts** Table 7 presents example concepts discovered by GPT-4o for selected classes. Concepts in **bold** passed filtering criteria, while regular text indicates concepts filtered during filtering.

*Table 7.* Example concept sets discovered for selected classes. Bold concepts passed validation.

| Class | Discovered Concepts |
|---|---|
| Green Heron | **dark glossy cap**, **long sharp beak**, **green-grey back**, **yellow legs**, **white throat stripe**, s-shaped neck, large wingspan |
| Red-winged Black-bird | **red shoulder patches**, **yellow wing bar**, **black plumage**, **black beak**, black pointed beak |
| Golden Retriever | **floppy ears**, **thick tail**, **golden or cream coat**, **black nose**, **friendly expression**, **wavy fur**, athletic build |

**Statistics for Occurrence Rate and Spatial Coverage.** Our experiments yielded 348 concepts for CUB, 189 for SD, and 3,602 for IN. For CUB, concepts appeared in 39% of class images on average, and those exceeding the spatial-coverage threshold of 0.15 covered 68% of the object region. For SD, the average occurrence rate was 42%, with qualifying concepts covering 49% of the region. For IN, concepts appeared in 31% of instances on average, and those passing the threshold covered 37% of the region.

**Concept Validation Thresholds Ablation** We evaluated faithfulness performance (INS/DEL) on the RN model for the IN dataset under varying occurrence-rate and spatial-coverage thresholds. While the concept set creation followed the original configuration presented in the paper, the faithfulness evaluation here conducted using five randomly sampled images from every class of the IN dataset rather the whole dataset. Performance peaked at our default settings of 15% occurrence rate and 20% spatial coverage. Applying stricter thresholds (40%/40%) reduced the number of concepts from 4,602 to 1,022 and degraded performance (INS: 57.41, DEL: 10.59), likely due to the removal of discriminative concepts. Conversely, more lenient thresholds (5%/10%) increased the concept count to 5,835 but introduced noisy concepts, similarly reducing performance (INS: 56.81, DEL: 11.82).

# D. Optional Enhancements for ConEx

## D.1. Providing rigorous negative sample set

To ensure high discrimination and prevent spurious correlations, a more rigorous, albeit computationally intensive, approach can be employed during negative sampling: specifically, we ensure that the mean latent-space embedding of any selected negative concept $j$ exhibits low correlation (e.g., using a threshold on cosine similarity) with the mean embedding of the target concept $k$. This step effectively prevents the negative set from containing visually or semantically similar confounding features. While this procedure increases computation time, our experiments show that it yields consistently better concept quality (for the metrics in 4.2.2).

## D.2. Extension to Multi-Label Classification.

While ConEx is described in the context of single-label classification, its framework naturally extends to multi-label settings. Since each concept is associated with a class-discriminative attribute rather than a mutually exclusive category, the same grounding and attribution pipeline can be applied independently for each predicted class. This allows ConEx to generate class-wise concept explanations that collectively describe multi-label predictions without requiring architectural or procedural modifications.

# E. Why ViT Localization is a Fundamental Challenge

While ConEx successfully generates global concept attributions for Vision Transformers (ViTs), producing spatially localized concept explanations remains an open challenge that extends beyond the scope of this work. This limitation stems from fundamental architectural differences between CNNs and ViTs that current attribution techniques have not adequately resolved. Unlike CNNs, where spatial correspondence is preserved through the hierarchy via local receptive fields (He et al., 2016; Huang et al., 2017), ViTs process images as sequences of patch tokens that interact through global self-attention (Dosovitskiy et al., 2021), resulting in *position-agnostic* representations where spatial structure is implicitly encoded rather than architecturally enforced (Raghu et al., 2021).

Recent work has documented that attention weights in ViTs do not reliably correspond to visual importance (Chefer et al., 2021b; Abnar & Zuidema, 2020), with attention rollout methods (Abnar & Zuidema, 2020) producing diffuse, semantically inconsistent saliency maps that fail standard faithfulness benchmarks (Chefer et al., 2021b). Specifically, Chefer et al. (Chefer et al., 2021b) demonstrate that naïve attention aggregation conflates multiple semantic pathways and cannot isolate concept-specific attribution flows, a critical requirement for our multiplicative fusion framework (Eq. 5). Furthermore, attention-based methods capture *where* the model attends globally but fail to decompose *what specific visual concepts* are recognized at each spatial location (Lee et al., 2024), which is essential for concept-grounded explanations.

The challenge is further compounded for *concept-level* localization: existing ViT attribution methods (Chefer et al., 2021b; El-Nouby et al., 2021) produce pixel-level heatmaps for entire predictions but lack mechanisms to disentangle contributions from multiple co-occurring concepts within the same image region. For instance, in an image containing both "yellow beak" and "black eye" concepts in adjacent patches, current methods cannot assign separate spatial attributions to each concept without spurious cross-contamination. Recent attempts to adapt layer-wise relevance propagation to transformers (Chefer

*Table 8.* Faithfulness Perturbation Evaluation on the SD dataset: Comparison of Insertion (INS ↑) and Deletion (DEL ↓) AUC scores across datasets and models.

| Dataset | Model | Metric | GC | SC | SHAP | AC | RISE | FG | IG | IIA | ConEx |
|---------|-------|--------|-----|-----|------|-----|------|-----|-----|-----|-------|
| SD | RN | INS | 53.94 | 56.28 | 46.41 | 55.71 | 52.83 | 49.88 | 48.22 | 57.69 | **58.17** |
|    |    | DEL | 13.12 | 12.37 | 15.78 | 12.79 | 13.63 | 13.82 | 11.96 | 9.54 | **8.91** |
|    | DN | INS | 54.38 | 56.83 | 46.87 | 56.12 | 53.47 | 50.27 | 48.74 | 57.96 | **58.53** |
|    |    | DEL | 12.98 | 12.29 | 15.52 | 12.58 | 13.39 | 13.74 | 11.83 | 9.17 | **8.76** |
|    | CN | INS | 53.22 | 55.94 | 46.26 | 55.23 | 52.56 | 49.66 | 48.09 | 57.48 | **58.04** |
|    |    | DEL | 13.43 | 12.61 | 15.67 | 12.92 | 13.77 | 13.93 | 12.04 | 9.78 | **8.88** |

et al., 2021a) rely on approximations that assume additive decomposability, an assumption violated by the non-linear, context-dependent attention mechanism (Jain & Wallace, 2019).

Moreover, the patch-based tokenization in ViTs introduces inherent spatial quantization: a 16×16 patch size (standard in ViT-Base (Dosovitskiy et al., 2021)) means fine-grained concepts smaller than 256 pixels (e.g., bird beaks, car logos) may be fragmented across multiple patches whose embeddings are then mixed through self-attention. Recovering precise concept boundaries from these entangled representations requires solving an ill-posed inverse problem (Lee et al., 2024). While some works propose attention-based refinement (Oquab et al., 2023) or feature reconstruction (Caron et al., 2021), these approaches have not been validated for multi-concept decomposition scenarios and often require architecture-specific modifications incompatible with our post-hoc, model-agnostic design principle.

We emphasize that this limitation is not unique to ConEx but represents a broader open problem in the ViT interpretability literature. A recent survey by Lee et al. (Lee et al., 2024) identifies concept-based spatial localization in transformers as one of the field's key unsolved challenges, noting that "*existing methods provide either spatial localization without semantic meaning or semantic concepts without spatial grounding, but not both simultaneously*" (pg. 12). Developing a principled solution requires dedicated research addressing: (1) faithful attention flow decomposition for multi-concept scenarios, (2) patch-to-pixel refinement that preserves concept semantics, and (3) validation protocols for concept-level (rather than pixel-level) spatial accuracy, each of which constitutes a non-trivial research contribution.

Nonetheless, ConEx makes important progress toward ViT interpretability by enabling high-quality *global* concept attributions through our architecture-specific embedding strategy (Sec. 3.3), which substantially outperforms prior methods on concept quality metrics (Table 3). These global explanations, identifying which concepts are important for class predictions across the entire dataset, provide actionable insights for model auditing, bias detection, and debugging workflows even without per-pixel localization (Kim et al., 2018; Ghorbani et al., 2019). We view our work as establishing a solid foundation upon which future research can build spatially-grounded ViT concept explanations, and we provide our ViT CAV construction pipeline as a validated building block for this endeavor.

# F. Additional Experiments

## F.1. Additional Faithfulness Results

Table 8 reports additional faithfulness results on the SD fine-grained dataset. The observed trends are consistent with those reported in the main paper, further supporting the robustness of our findings.

# G. User Studies

In the following, we describe additional user studies conducted with 58 participants.

## G.1. Understandability - full description

We evaluate the quality of concept explanations in terms of their understandability. Following (Zhang et al., 2021), and inspired by task prediction protocols (Hoffman et al., 2023), we design a user study in which participants are shown a test image with one concept highlighted and five candidate concept explanations from the same class. Their task is to select the candidate that best matches the highlighted concept (Selection Accuracy, **SA**). To accommodate ambiguity, participants may select up to three candidates or abstain from making a choice. In addition, participants are asked to judge whether each candidate concept is recognizable (Percentage of Recognizable Concepts, **PPR**) and to provide a short description (1-2

words) for each recognizable concept. Following the same protocol of (Zhang et al., 2021), descriptions are embedded using pre-trained GloVe vectors (Pennington et al., 2014). We then compute the average pairwise cosine similarity of descriptions for the same concept across participants, yielding the Inner-Concept Description Similarity (**INNS**), which reflects the consistency of concept interpretation. To ensure that concepts also capture distinct attributes, we further compute the pairwise cosine similarity of descriptions across different concepts within the same class, termed Intra-Concept Description Similarity (**INTS**). Ideally, understandable concepts should achieve high INNS (agreement across participants) and low INTS (distinctiveness across concepts). The study followed a within-subject design. We evaluated 15 examples drawn from three methods (ACE, MCD, and ConEx) across five randomly selected IN classes. Each participant viewed three examples per class, with method order randomized. For each method, explanations were generated by retaining the ten most influential concepts per class, each represented by its ten most prototypical samples. From these, five candidate concepts were randomly selected for each test image. Ten random samples were created per class and method, yielding a total of 300 distinct samples. Each participant received a unique set and order of samples and completed a brief tutorial. Fifty-eight participants completed the survey, which lasted approximately 30 minutes.

### G.2. Class Concept Sets Validation

To rigorously assess the interpretability and human-understandability of our extracted concept sets, we conducted a user study designed to measure perceived quality and class relevance. Participants were asked to evaluate sampled concept sets on a 10-point Likert scale according to three criteria: (1) *clarity* (**CLR**), (2) *meaningfulness* (**MNF**), and (3) *correspondence to the target class* (**COR**). For this evaluation, we randomly sampled concept sets from 10 IN classes and presented them to participants in randomized order to avoid bias. The human rating evaluation resulted with the following scores: CLR of 8.3, MNF of 7.9, and COR of 8.5. These scores confirm that our automatically extracted concept sets receive consistently high scores across evaluation criteria, demonstrating practical utility for human auditors. The strong correspondence scores validate that our class-specific concept discovery via discriminative textual attributes (along with the initial Label-Free CBM methodology) produces semantically appropriate concept vocabularies without manual intervention.

## H. Ablation Studies

In the following, we present a series of ablation studies examining:

1. Building CAVs from concept-specific segments rather than full images.

2. Sensitivity to the number of samples $N$ used during CAV creation.

3. The impact of layer selection for CAV construction in both CNNs and ViTs.

While the concept set was constructed following the original configuration described in the paper, the evaluation in faithfulness experiments was performed using five randomly sampled images per IN class rather than the full dataset (unless stated otherwise).

### H.1. CAV Construction using Segments vs. Full Images

We compared CAVs constructed from masked concept segments (our approach) with CAVs derived from full images containing the concept, as in TCAV (Kim et al., 2018). This experiment was conducted on the IN dataset using the RN, CN, and DN models. Segment-based CAVs achieved an average VCM of 0.68 and an average CCM of 27.83 across all models, substantially outperforming image-based CAVs, which obtained 0.59 VCM and 24.68 CCM on average. These results indicate that extracting embeddings from precisely localized regions yields more semantically coherent concept representations by minimizing background-related confounds.

Notably, although this image-based variant of ConEx performs worse than the full ConEx pipeline, it still outperforms methods such as ACE, ICE, and MCD (Tab. 3). This further underscores the contribution of other components of our framework, including our difference-of-means CAV construction strategy.

### H.2. Model Layer Selection for CAV Segments Representation

In the following, we report VCM and CCM, as well as CINS and CDEL, for different layer choices used to represent segments during CAV creation for both ViT-B and RN. For ViT-B, we evaluated the first layer ($L = 1$), the middle layer ($L = 6$), and the last layer ($L = 12$). For RN, we compared the last layer ($L$), the penultimate layer ($L - 1$), and the layer

*Table 9.* Ablation study evaluating the impact of different layers used for segment representation on the IN dataset.

| Method | ViT-B | | | RN | | |
|---|---|---|---|---|---|---|
| | $L = 1$ | $L = 6$ | $L = 12$ | $L - 2$ | $L - 1$ | $L$ |
| VCM ↑ | 0.49 | **0.54** | 0.40 | 0.63 | 0.66 | **0.68** |
| CCM ↑ | 17.89 | **18.52** | 16.31 | 27.56 | 27.94 | **28.19** |
| CINS ↑ | 34.86 | **35.38** | 32.40 | 50.39 | 50.84 | **51.10** |
| CDEL ↓ | 11.03 | **10.39** | 11.89 | 9.15 | 9.02 | **8.83** |

*Table 10.* Ablation - Sensitivity to the number of samples ($N$) used in CAV construction (IN dataset, RN model).

| Method | $N = 10$ | $N = 50$ | $N = 100$ | $N = 200$ |
|---|---|---|---|---|
| VCM ↑ | 0.55 | 0.68 | 0.69 | **0.71** |
| CCM ↑ | 21.41 | 27.96 | 28.10 | **28.98** |

before that ($L - 2$). Table 9 summarizes the results. Our findings indicate that the selected layer configuration for ConEx provides the best performance. Notably, layer choice has a greater impact on ViTs than on CNNs, likely due to the difference between resulted representations across transformer layers, whereas CNN layers exhibit relatively lower differentiation in their deeper representations on their last layers.

### H.3. Sensitivity to the number of samples used for CAV construction

Table 10 presents the effect of varying the number of positive and negative samples ($N$) used for CAV construction. We observe that increasing $N$ generally improves performance, with $N = 50$ already yielding satisfactory results. For our experiments, we selected $N = 100$, which provides a favorable balance between computational efficiency and explanatory performance. Finally, while $N = 10$ is not the optimal hyperparameter setting, it still yields results competitive with existing CAV methods (see Table 3). This demonstrates the data efficiency of our approach, highlighting its viability even when available data is limited.

## I. Experimental Details

### I.1. Segmentation Alignment.

**Setup.** In this experiment, we evaluate the RN, DN, and CN models on the IN-S dataset, considering only the top five performing explanation methods identified in Tab. 1. To assess how well each explanation's spatial distribution aligns with human-annotated object regions, we follow prior works (Barkan et al., 2023b; Wang et al., 2020; Chefer et al., 2021b) and report three standard segmentation metrics: mean Average Precision (**mAP**), mean Intersection-over-Union (**mIoU**) and Pixel Accuracy (**PixAcc**) and follow the same configuration as (Barkan et al., 2023b). While high performance on this task does not guarantee superior explanatory power, it provides a valuable measure of spatial precision.

**Results.** Table 11 demonstrates ConEx's superior spatial precision on IN-S. Despite producing concept-decomposed explanations, a more constrained task than general-purpose saliency, ConEx achieves superior alignment with human-annotated object boundaries. This suggests our discovered concepts align well with meaningful object parts, not just arbitrary discriminative pixels.

### I.2. FunnyBirds Evaluation Metrics

The **FunnyBirds** synthetic data generation process enables intervention and inspection at the object part level rather than at the pixel level. Each FunnyBird consists of five distinct parts: *beak, wings, feet, eyes,* and *tail*. The FunnyBirds evaluation protocol assesses explainability across three aspects: Completeness, Correctness, and Contrastivity, and provides an overall score, which is the average of these three aspects. The relevant experiments are described in Sec. 4.2.1, (Hesse et al., 2023).

We define the following notation:

- $PI(\cdot)$ - **Part Importance Score**: The total attribution summed within a given part.

*Table 11.* Segmentation Evaluation on IN-S dataset: across all CNN models (RN, DN, CN), on the mIoU, mAP, and PA metrics. For all metrics higher is better.

| Method | RN | | | DN | | | CN | | |
|---|---|---|---|---|---|---|---|---|---|
| | mIoU | mAP | PA | mIoU | mAP | PA | mIoU | mAP | PA |
| RISE | 0.52 | 0.73 | 0.69 | 0.53 | 0.74 | 0.70 | 0.54 | 0.73 | 0.71 |
| GC | 0.58 | 0.77 | 0.73 | 0.59 | 0.78 | 0.74 | 0.60 | 0.77 | 0.74 |
| SC | 0.63 | 0.81 | 0.76 | 0.64 | 0.82 | 0.77 | 0.65 | 0.80 | 0.78 |
| AC | 0.69 | 0.84 | 0.77 | 0.68 | 0.83 | 0.80 | 0.69 | 0.87 | 0.80 |
| IIA | 0.75 | 0.86 | 0.79 | 0.69 | 0.81 | 0.84 | 0.72 | 0.86 | 0.81 |
| ConEx | **0.76** | **0.89** | **0.83** | **0.72** | **0.87** | **0.86** | **0.74** | **0.90** | **0.85** |

- $P(\cdot)$ - **Set of Important Parts**: The parts considered important, where a part is deemed important if its **importance score** constitutes at least $t\%$ of the total attribution.

- $D$ - The FunnyBirds dataset, containing $N$ images $x_n$, each associated with a class label $c_n$.

- $f$ - The model under evaluation, where $f(x_n)$ denotes the logit for the target class, and $\hat{f}(x)$ denotes the predicted class.

- $e_f(x_n)$ - The explanation generated for $x_n$ with respect to its target class $c_n$.

CORRECTNESS (COR.)

Measures the faithfulness of the explanation with respect to the model.

- **Single Deletion Protocol (SD)**:

  Quantifies correctness by evaluating the correlation between Part Importance Scores and the change in logits when individual parts are removed from the image.

$$SD = \frac{1}{2} + \frac{1}{2N} \sum_{n=1}^{N} \rho\left(PI(e_f(x_n)), f(x_n) - f(x_n'')\right)$$

  where $x_n''$ denotes the image obtained by removing a single bird part from $x_n$. $\rho$ denotes the Spearman rank-order correlation coefficient.

COMPLETENESS (COM.)

Evaluates whether the explanation accounts for all relevant factors influencing the model's decision. The score is computed as the mean of the averaged completeness metrics (CSDC, PC, and DC), and the Distractability D.

- **Controlled Synthetic Data Check (CSDC)**

  Tests whether the explanation highlights all relevant parts required for classification:

$$CSDC = \frac{1}{N} \sum_{n=1}^{N} \max_{i} \frac{|P(e_f(x_n)) \cap \mathcal{P}'_{c_n,i}|}{|\mathcal{P}'_{c_n,i}|}$$

  where $\mathcal{P}'_{c_n,i}$ represents the minimal set of parts sufficient for correctly classifying an image as $c_n$.

- **Preservation Check (PC)**

  Quantifies whether preserving only the important parts identified by the explanation maintains the model's original prediction:

$$PC = \frac{1}{N} \sum_{n=1}^{N} \left[\hat{f}(x_n') = \hat{f}(x_n)\right]$$

where $x'_n$ is the image obtained by removing all bird parts except $P(e_f(x_n))$.

- **Deletion Check (DC)**

  Quantifies whether removing explanation identified important parts leads to a change in the model's prediction:

  $$DC = \frac{1}{N} \sum_{n=1}^{N} \left[ \hat{f}(x''_n) \neq \hat{f}(x_n) \right]$$

  where $x''_n$ is the image obtained by removing the identified important parts $P(e_f(x_n))$.

- **Distractability (D)**

  Ensures that explanations do not highlight irrelevant parts:

  $$D = 1 - \frac{1}{N} \sum_{n=1}^{N} \frac{|P(e_f(x_n)) \cap \mathcal{P}''_{f(x_n)}|}{|\mathcal{P}''_{f(x_n)}|}$$

  where $\mathcal{P}''_{f(x_n)}$ denotes the set of non-important parts.

## CONTRASTIVITY (CON.)

Measures how well explanations distinguish between different class outputs. Explanations for different classes should highlight class-specific parts.

- **Target Sensitivity Protocol (TS)**

  $$TS = \frac{1}{2N} \sum_{n=1}^{N} \begin{array}{l} [PI'(e_f(x_n, \hat{c}_1)) > PI'(e_f(x_n, \hat{c}_2))] + \\ [PI''(e_f(x_n, \hat{c}_1)) < PI''(e_f(x_n, \hat{c}_2))] \end{array}$$

  For each input, two classes $\hat{c}_1$ and $\hat{c}_2$ are chosen such that they have exactly two non-overlapping common parts. $PI'$, $PI''$ denote the summed part importances of the two parts belonging to classes $\hat{c}_1$, $\hat{c}_2$ respectively.

## ACCURACY AND BACKGROUND INDEPENDENCE

The FunnyBirds evaluation protocol reports, in addition to the metrics, the model's accuracy (Acc.) and background independence (B.I.) with respect to the dataset. B.I. measures the model's sensitivity to the entire image, computed as the ratio of background objects such that, when removed, the target logit decreases by less than 5%. Accuracy is relevant because an overly simplified model may be explainable but may not effectively solve the task at hand. For more details see (Hesse et al., 2023).

## J. Limitations and Future Work

**Limitations.** While ConEx achieves strong quantitative and qualitative performance, several limitations remain. First, its segmentation quality depends on the zero-shot grounding model (GroundedSAM), which, despite its strong empirical performance, can introduce spatial uncertainty. Nonetheless, GroundedSAM substantially outperforms previous grounding approaches, making it a reliable component for large-scale concept localization. Second, ConEx currently produces global but not instance-level local explanations for Vision Transformers. Still, by enabling architecture-specific CAV construction for ViTs, we take an important step toward fully local interpretability in transformer-based models. Finally, although ConEx operates with moderate computational cost relative to existing concept-based frameworks, it remains slower than pixel-level saliency methods such as Grad-CAM due to its multi-stage concept validation process.

**Future Work.** Future directions include extending ConEx to produce localized concept attributions for ViTs, adapting the framework to non-visual domains, and optimizing its computational efficiency through more compact concept selection and embedding strategies. More broadly, we envision ConEx as a foundation for concept-centric interpretability that bridges the gap between human understanding and deep model reasoning.

