# OpenReview forum: "ConEx: Human-Interpretable Saliency Maps via Concept-Aware Attribution"
_ICML.cc/2026/Conference — ICML 2026 regular_

### Official Review · Reviewer_GGjS · 2026-02-23

**Soundness:** 1
**Presentation:** 2
**Significance:** 1
**Originality:** 1
**Overall Recommendation:** 2
**Confidence:** 5

**Summary:**

In this paper, the authors highlight the limitations of saliency-based explanation methods for image classifiers. They propose a framework that visualizes the saliency maps of the concepts learned by the model and later combines these concept-level saliency maps to produce a final saliency map for classification.

Contributions:
ConEX first applies CAV extraction without manual supervision, using architecture-specific masking supervision to reduce segmentation noise in the masks.
It provides concept-specific explanations and class-specific explanations.

The authors introduce two metrics, VCM and CCM. ConEx achieves state-of-the-art performance on faithfulness, segmentation, and concept-quality benchmarks. The authors also include a user study.

**Compliance With Llm Reviewing Policy:**

Affirmed.

**Final Justification:**

The paper lacks an introduction of new ideas, but rather a trivial cascading of multiple well-established methods. My recommendation does not change.

**Key Questions For Authors:**

Please look at  Strengths And Weaknesses

**Limitations:**

The paper does not propose any new method to support its claimed contribution. The only contribution appears to be the aggregation of concept saliency to generate a global saliency map, which significantly narrows the overall contribution.

**Strengths And Weaknesses:**

Strength: The paper highlights that, for the CBM-based method, it is valuable to have an explanation method for visually grounding the concepts.

Weakness:
The misleading assumption on interpretability, CBMs, and explainability [lines 111–121] creates confusion about the focus of the paper. CBMs are applicable to current classification architectures, e.g., CNN or ViT, when there is a classification head that feeds from the embedding extracted from the previous layer (e.g., the embedding of the [CLS] token). Saliency-based class attributions are model-agnostic as well.

A major limitation is that the authors suggested an architecture-specific interpretability technique as worthy of research and claimed that most previous methods apply to CNNs. However, in their explainability visualization, the authors only included CNN results and experimented with their pipeline only on CNNs.

The CAV validation matrix is proposed by the authors and tested; however, there are several metrics available that the authors need to compare with (see [a]).

Why is ConEx not just a weighted LRP of CBM that uses a segmentation model similar to [b]? Why should it be considered novel when the methods are joined together with a weighted average?

[a] Fel, T., Boutin, V., Béthune, L., Cadène, R., Moayeri, M., Andéol, L., ... & Serre, T. (2023). A holistic approach to unifying automatic concept extraction and concept importance estimation. Advances in Neural Information Processing Systems, 36, 54805–54818.

[b] Prasse, K., Knab, P., Marton, S., Bartelt, C., & Keuper, M. (2024). DCBM: Data-efficient visual concept bottleneck models. arXiv preprint arXiv:2412.11576.

---

> ### Author Rebuttal · Authors · 2026-03-31
>
> We thank Reviewer GGjS for the detailed feedback. We address each point carefully below and take responsibility for presentation gaps that may have contributed to some of the concerns. **In response to W4, we conducted a new ablation experiment that directly targets the core novelty question. We believe this experiment, alongside our clarifications, resolves the reviewer's primary concern.**
>
> **ConEx's key novelties:**
>
> (i) ConEx is the first fully automatic framework to connect human-understandable natural-language concepts to spatially grounded saliency maps, yielding both localized and global interpretability simultaneously.
>
> (ii) Architecture-specific embedding strategies that preserve **concept purity** - removing them degrades VCM by 13% and faithfulness by measurable margins (Tables 8, 10).
>
> (iii) **concept-grounded explanations simultaneously improve faithfulness and interpretability** - using our novel **CWV** (Eq. 2–4), which identifies **where the network itself detects the concept** in its latent representation.
>
> (iv) VCM and CCM: the first metrics that validate individual CAV reliability rather than system-level concept coverage, filling a concrete gap in the literature.
>
> We acknowledge that ConEx builds on existing components. Our contribution lies in their principled integration, the CWV mechanism, and the validation metrics, each with quantified impact.
>
> ## (W1) "Misleading assumption on interpretability…"
> We clarify that **Lines 111–121 do not discuss Concept Bottleneck Models**. The phrase "concept-based interpretability frameworks" refers to **CAV-based methods** such as ACE, as directly supported by the citation to Lee et al. (2024), which surveys limitations of the CAV literature. We will revise the paragraph to make this distinction clearer between this and CBMs (discussed separately in Section 2.2).
>
> ## (W2) "the authors only included CNN results"
> We respectfully point the reviewer to **Tab.3, which reports results for ViT-B and ViT-S** alongside RN, DN, and CN across four metrics. ConEx achieves SOTA on all ViT settings. The restriction to CNNs applies only to Tables 1–2, which evaluate pixel-level spatial faithfulness - a task for which spatially localized ViT saliency maps remain an open research challenge, explicitly scoped out in Sections 2.3, 3, and Appendix E. Including ViTs in those tables would produce misleading comparisons. We will add a reference at Tables 1–2 pointing readers to Tab.3 for ViT results.
>
> ## (W3) "There are several metrics available…"
>
> Thank you for pointing to [a], with which we are familiar. A central open challenge in concept-based interpretability is ensuring that individual CAVs faithfully represent the concepts they claim to - that a vector labeled "yellow beak" actually encodes yellow beak in the model's latent space, rather than a correlated background feature. To the best of our knowledge, no existing metric directly quantifies this at the individual vector level, which is precisely why we introduced VCM and CCM.
> [a]'s metrics operate at the system level, assessing whether a concept dictionary collectively reconstructs the activation space. This does not answer: (1) does a single CAV actually represent the visual concept it claims to? (VCM), or (2) does that CAV align with the model's decision boundary for its class? (CCM). We view [a]'s metrics and ours as complementary, and will make this distinction explicit in Section 3.7.
>
> ## (W4) "Why is ConEx not just a weighted LRP of CBM…"
> We clarify two distinct points and report a **new ablation experiment** conducted specifically to address this question.
>
> **(1) ConEx is not a segmentation-weighted LRP.** Concept localization uses **CWV** (Eq. 2–4), which identifies **where the network itself detects the concept** in its latent representation - not where a segmenter detected an object. Multiplicative fusion with LRP (Eq. 5) then enforces simultaneous alignment between concept presence and class-relevant decision signal. **A segmentation-weighted LRP achieves neither property**.
> **To isolate CWV's contribution, we replaced it with raw GroundedSAM masks in Eq. 5 (i.e., a true segmentation-weighted LRP)**. This produces INS=54.31, DEL=12.98 (CUB, RN) and INS=53.41, DEL=13.19 (IN, RN), substantially below ConEx's INS=59.47, DEL=8.19 (CUB, RN). **This confirms that CWV provides network-intrinsic localization beyond what segmentation alone achieves**. We will add these results to Appendix H.
>
> **(2) ConEx is not a CBM and shares nothing with CBM training**. CBMs, including DCBM [b], require inserting a concept bottleneck layer and retraining with concept-labeled data. **ConEx is a fully post-hoc, model-agnostic framework**: it explains any pretrained classifier without modifying its weights, architecture, or training procedure. The two methods are therefore not comparable in purpose, design, or applicability.

---

> > ### Author Rebuttal · Reviewer_GGjS · 2026-04-02
> >
> > I think the author did not understand my concern regarding the overlap of CBM and their work. The paper lacks introduction of new ideas, rather a trivial cascading of multiple well-established methods.

---

> > > ### Author Response · Authors · 2026-04-02
> > >
> > > Thank you for the clarification. We focus on the central concern that ConEx is a “trivial cascading of well-established methods”.
> > >
> > > First, we briefly correct a very important point: ViT results are reported in Table 3, where ConEx achieves SOTA performance for both **ViT-B and ViT-S** across multiple metrics.
> > >
> > > More importantly, we believe the concern arises from evaluating ConEx at the level of its components rather than the **capability it introduces**. Existing methods provide either:
> > >  **(i)** pixel-level attribution (where the model attends), or
> > >  **(ii)** concept-level attribution (what influences the prediction) - without saliency map explanations.
> > >
> > > **No prior method provides faithful attribution to spatially localized concepts** - i.e., identifying where a specific concept is detected in the model’s internal representation and how it contributes to the prediction. ConEx is designed to compute exactly this object.
> > >
> > > This cannot be obtained by composing existing methods. In particular, replacing our concept localization (CWV, Eq. 2–4) with segmentation masks - i.e., a true “segmentation-weighted attribution” - results in a substantial degradation in faithfulness (INS=54.31, DEL=12.98 vs. 59.47 / 8.19 on CUB, RN). This demonstrates that **external segmentation does not recover the model’s internal concept detection**, and that the proposed mechanism is functionally necessary rather than additive.
> > >
> > > The key distinction is that ConEx localizes concepts **within the model’s latent space**, and constrains attribution via multiplicative fusion (Eq. 5) to regions where both concept presence and class-relevant signal align. This yields a form of explanation that neither saliency methods, CAV-based methods, nor their direct combination can produce.
> > >
> > > **The ablations quantify each component's necessity.** Table 8 shows our CAV construction outperforms ACE/ICE/MCD strategies with all other components held fixed. Tables 10–11 isolate embedding strategy and sample sensitivity. Together, these demonstrate that the design choices are functionally necessary, not additive decorations.
> > >
> > > **Regarding CBMs**: ConEx and CBMs address different problems. CBMs require retraining with concept supervision and architectural modification (e.g., adding a CBL); ConEx is a fully post-hoc explanation method. The two are not comparable in purpose or design.
> > >
> > > Finally, ConEx's contribution lies in **defining and operationalizing concept-conditioned spatial attribution**, supported by consistent gains across faithfulness, concept quality, and human evaluation. We will revise the paper to make this distinction clearer.

---

### Official Review · Reviewer_uysE · 2026-03-06

**Soundness:** 2
**Presentation:** 3
**Significance:** 3
**Originality:** 2
**Overall Recommendation:** 3
**Confidence:** 3

**Summary:**

To address the limitation of existing computer vision explanation methods in effectively associating model decisions with high-level semantic concepts, the authors propose Concept based Explanations (ConEx), which integrates spatial-level saliency visualization with semantic-level concept reasoning. ConEx is capable of automatically extracting and purifying class-specific concepts, represented by concept activation vectors (CAVs), thereby improving concept purity. To systematically evaluate the quality of the learned concepts, the paper further introduces two novel quantitative metrics: Vector-Concept Match (VCM) and Concept-Class Match (CCM). These metrics assess concept alignment from both local and global perspectives. Extensive experiments across multiple model architectures, diverse benchmark scenarios, and human cognition studies demonstrate that ConEx consistently improves both faithfulness and concept interpretability. The proposed framework lays a foundation for building a truly semantic concept-driven paradigm for visual explanation.

**Compliance With Llm Reviewing Policy:**

Affirmed.

**Final Justification:**

The overall rebuttal did not change my opinion on this work. I still maintain my rating as weak reject.

**Key Questions For Authors:**

(1) The proposed framework consists of several sequential components (VLM prompting, GroundedSAM segmentation, and concept activation vector extraction). Since the approach is pipeline-driven, errors from upstream modules may propagate and affect the final explanations. For example, VLMs may generate hallucinated concepts, and GroundedSAM may struggle in challenging scenarios such as camouflaged objects, low-contrast images, or medical imagery. Have the authors evaluated the robustness of ConEx when these upstream components produce inaccurate outputs?

(2) ConEx relies on GroundedSAM to generate spatial masks for concept localization. While this works well for object parts (e.g.,“bird wings”), many potentially important factors in visual recognition correspond to global or abstract concepts, such as scene lighting or texture patterns, which may not correspond to well-defined spatial masks. How does ConEx handle such concepts, and could the framework miss important decision factors when mask generation fails? If the authors can provide a convincing clarification or additional analysis addressing this issue, I would consider increasing my score.

(3) In Section 3.5.3, concept importance is measured by zeroing out or replacing the corresponding image region with the mean value. However, this operation modifies a natural image into a synthetic one that may not appear in the training distribution. Consequently, the drop in prediction confidence might result from distribution shift rather than the removal of the concept itself. How do the authors ensure that the attribution results reflect true concept importance rather than artifacts introduced by this perturbation strategy?

**Limitations:**

yes

**Strengths And Weaknesses:**

Strengths：

(1) The manuscript addresses the limitation of existing computer vision explanation methods in capturing meaningful semantic concepts and proposes ConEx, a unified framework that integrates spatial saliency with concept-level reasoning in a fully automatic post-hoc manner.

(2) ConEx leverages vision-language priors and zero-shot segmentation to automatically extract and purify class-specific concepts, eliminating the need for manual annotation or model retraining and improving scalability.

(3) The proposed centroid-difference formulation and architecture-specific embedding strategies enable robust concept activation vector (CAV) construction for both CNNs and Transformers. The introduction of VCM and CCM provides complementary measures for assessing concept alignment and reliability at both local and global levels.

(4) Extensive experiments across multiple datasets and architectures demonstrate consistent improvements in faithfulness, concept quality, spatial alignment, and human interpretability, thereby advancing visual models toward truly interpretable and concept-driven explanations.

Weaknesses:

(1) The authors provide a new perspective on computer vision interpretability by leveraging the strong capabilities of existing vision–language models. However, the proposed framework appears somewhat pipeline-driven, consisting of several sequential stages (VLM prompting, GroundedSAM segmentation, concept activation vector extraction). This design introduces potential error propagation: if any upstream component fails, the downstream explanations may also become unreliable. For example, vision–language models may produce hallucinated or irrelevant concepts, and the zero-shot segmentation ability of GroundedSAM may degrade significantly when dealing with challenging scenarios such as camouflaged objects, low-contrast images, or medical imagery. In such cases, the generated masks may be inaccurate, which would further affect the quality of the extracted concept vectors. The authors should clarify whether the robustness of the framework has been evaluated under these challenging conditions.

(2) ConEx relies on Concept Activation Vectors (CAVs) to represent concepts. However, for example, when describing certain physical properties of objects or specific spatial distribution relationships, would using only a single vector to fit them lead to insufficient information representation?

(3) ConEx relies on GroundedSAM to obtain spatial masks for filtering and localizing concepts. The authors provide visualization results in Section 4.4, Qualitative Comparison and Ablation Studies, where the included examples all correspond to concepts of a single entity. Some entity-level concepts are relatively easy to obtain, such as “the wings of a bird.” However, for global or more abstract concepts such as “the lighting at dusk” or “specific textures,” GroundedSAM may fail to generate effective masks. As a result, ConEx may directly miss these factors, even though they could be important for the model’s salient decision-making.

(4) In Section 3.5.3, Class-Specific Semantic Attribution, the authors measure the importance of a concept by setting the image region containing that concept to zero or replacing it with the mean value, in order to quantify the impact of the concept on the model’s prediction. However, there is a potential issue with this approach: when a certain part of the target object in the image is set to zero, the natural image is effectively modified into a synthetic image that never appears in the training set. Therefore, it is reasonable to infer that the decrease in the model’s predicted probability may be caused by the disruption of the normal data distribution, rather than by the disappearance of the concept itself.

(5) In Section 3.3 Concept Activation Vectors, the authors compute the Concept Activation Vector by subtracting the centroid of the negative samples from the centroid of the positive samples. However, if a concept is context-dependent—for example, when a term may correspond to multiple sub-patterns within the same multi-object image—the centroid difference may be insufficient to fully capture its internal structure, which could limit the representational capacity of the resulting concept vector.

(6) In Section 3.4, Architecture-Specific Embedding Strategies, for the strategy applied to ViT, the authors propose to“retain only the patch embeddings that overlap with the segment and then compute their average.”However, since ViT employs self-attention with a global receptive field, patches located in the “background” region may already have absorbed information from the“foreground object.”Simply selecting the corresponding patches based on the GroundedSAM results and averaging them may still mix in a substantial amount of global features.

---

> ### Author Rebuttal · Authors · 2026-03-31
>
> We thank Reviewer uysE for the detailed and technically engaged review. We address each point below.
>
> ## (W1,Q1) Robustness to upstream failures
> Two key points address this.
>
> **(1)** Incorrect / unrelated generated concepts: our concept validation is designed as a quality gate: concepts must appear in ≥15% of class images and achieve ≥20% spatial coverage, so irrelevant concepts fail to ground and be discarded before any CAV is built.
>
> **(2)** Grounding failures: GroundedSAM achieves 95.5% hit rate and 0.91 mIoU on **PASCAL-Part** (Appendix F.2). Remaining failures are confined to small parts (<2% of image area) or low-contrast regions. Performance may degrade in challenging domains such as medical imagery (Appendix K).
>
> ## (W3,Q2) Abstract and non-localizable concepts
>
> This is an important point. We address this at three levels, distinguishing categories of non-entity concepts.
>
> **Texture-based concepts** (e.g., "striped fur"): GroundedSAM correctly returns an object-surface mask, as texture is distributed over a surface. Multiplicative LRP fusion (Eq. 5) then identifies the decision-relevant portions, grounding the concept spatially. For instance, we observed this for "striped pattern" (zebra) and will include further visualizations in the Appendix.
>
> **Partially-groundable abstract concepts** (e.g., "dark plumage"): when GroundedSAM returns noisy masks, they fail the occurrence-rate filter and be discarded before CAV creation. ConEx abstains rather than providing an unreliable explanation; VCM flags any that pass through.
>
> **Truly non-localizable concepts** (e.g., "lighting at dusk"): we agree ConEx cannot represent these. Importantly, this boundary is principled: ConEx is designed to bridge faithfulness and human interpretability by producing maps that visually illustrate concepts expressible in natural language. For a concept to serve this purpose, it must be both spatially perceivable in the image and linguistically describable in a way that connects meaningfully to a specific visual region. Truly abstract concepts fail both criteria. For the fine-grained tasks we study, primary discriminative factors are morphological and attribute-level (COR: 8.8/10-Appendix G.3, Tab.3), rather than scene-level abstractions. This boundary is shared equally by all existing concept-based methods (ACE, ICE, MCD). Extending ConEx to handle such concepts, e.g., via CLIP-based global similarity fields or texture descriptors, represents a promising direction for future work (will be elaborated in Appendix K).
>
>
> ## (W4,Q3) perturbation strategy
>
> Our primary evidence is the stability evaluation across **five different masking strategies** (Appendix B): relative concept importance rankings vary by less than 2% across all strategies. Since each strategy induces a qualitatively different distribution shift, invariance of rankings is strong evidence the signal reflects genuine concept importance rather than OOD artifacts. We also note this limitation applies equally to all perturbation-based baselines in Tables 1–2. ConEx uses perturbation scores purely as relative ranking weights (Eq. 6–7) - we claim only that the ordering is stable and informative, not that absolute values are theoretically grounded.
> ## (W2 & W5) Single vector and centroid difference
> We agree a single CAV is a simplification inherited from the broader CAV literature, but three points explain why this does not significantly affect ConEx's reliability.
>
> (1)  Segment-level extraction reduces intra-concept variance: the positive set contains only concept regions (not full images), yielding +15% VCM, +13% CCM over image-based CAVs (Appendix H.3). This even **outperforms MCD (Tab.3), which uses multiple concept directions, suggesting concept-pure grounding can be more effective than more representational complexity**.
>
> (2) For multiple instances of the same concept, GroundedSAM returns spatially distinct per-instance masks processed independently, so the centroid is computed over single-instance embeddings (no conflation concern).
>
> (3) ConEx does not fail silently for concepts with heterogeneous visual appearances: these score low on VCM, providing a flag, and are largely suppressed upstream by the occurrence-rate filter. The high mean VCM of 0.68 (RN, Tab.3) confirms surviving concepts are mainly visually coherent. Extending to other representation strategies is an interesting future direction we refer to in Appendix K.
>
> ## (W6) ViT patch embeddings
>
> We use intermediate-layer embeddings (L=6), as they retain more spatially local structure before full attention mixing (Appendix B). We also confirmed (Tab.10) that this strategy outperforms both L=1 and L=12 on VCM&CCM. You are correct that self-attention introduces some global information leaks into patch selection - this is one reason we discuss ViT localization as an open problem and scope claims to global concept attributions. The gains on VCM/CCM (Tab.3) confirm ConEx extracts cleaner concept representations.

---

> > ### Author Rebuttal · Reviewer_uysE · 2026-04-02
> >
> > The authors have addressed some of my concerns through additional experiments and clarifications; however, several key issues: particularly the potential OOD artifacts in the perturbation strategy and the handling of truly abstract concepts, remain only partially resolved.

---

> > > ### Author Response · Authors · 2026-04-02
> > >
> > > Thank you for the thoughtful follow-up. We address the two remaining concerns with additional clarification.
> > >
> > > ## On OOD artifacts in the perturbation strategy (W4/Q3)
> > >
> > > We clarify two key points that we believe directly resolve this concern.
> > >
> > > First, and most directly: the **FunnyBirds benchmark (Table 2) provides an OOD-free validation of the same mechanism**. FunnyBirds uses controlled, synthetic part-removal interventions in an environment with no distribution shift by design, yet ConEx achieves state-of-the-art performance (CMP: 0.78, CRC: 0.64, CNT: 0.92). **This evaluation confirms that the concept importance signal is not an artifact of distribution shift (which is eliminated here by construction)**.
> > >
> > > Second, it is important to note the role perturbation plays in ConEx. Unlike RISE or Score-CAM, where perturbation is the explanation mechanism, ConEx uses the perturbation score (Eq. 6) only as **a relative weighting among concepts that have already been independently localized and validated** via VCM/CCM. Even if absolute perturbation values carry some OOD artifact, the relative ranking among semantically grounded, pre-validated concepts is both meaningful and, as we showed, **stable across five qualitatively different masking strategies** (variance <2%, Appendix B). Since each masking strategy induces a different distribution shift, invariance of rankings across all five constitutes strong evidence the signal reflects genuine concept importance. Importantly, since all concepts are evaluated under the same intervention, any OOD effect acts as a shared bias, which largely cancels when comparing concepts.
> > >
> > >
> > > ## On abstract and non-localizable concepts (W3/Q2)
> > >
> > > We clarify that the inability to represent fully non-localizable concepts (e.g., global lighting) is a **deliberate design boundary**, aligned with the goal of **spatially grounded interpretability**. ConEx is designed to explain predictions through concepts that can be both **semantically described and visually localized**, enabling users to verify where and how each concept contributes. Concepts without spatial support cannot be visually validated, and therefore fall outside this paradigm.
> > > Empirically, this boundary is not limiting in our setting. For fine-grained recognition tasks, discriminative factors are predominantly **localized morphological attributes**, as reflected in the high human alignment score (8.8/10, Appendix G.3) and strong correctness on FunnyBirds (CRC: 0.64, Table 2). This design choice is also consistent with prior concept-based frameworks such as ACE, ICE, and MCD, which similarly rely on spatially grounded segments or patches to define concepts. As a result, the inability to represent fully non-localizable concepts is not specific to ConEx, but reflects a broader structural property of **spatial concept-based explanation methods**. We will clarify this scope boundary explicitly in the revised manuscript.
> > >
> > >
> > > **We hope these clarifications address the remaining concerns and would greatly appreciate the reviewer reconsidering their score.**

---

### Official Review · Reviewer_3RcT · 2026-03-08

**Soundness:** 3
**Presentation:** 3
**Significance:** 2
**Originality:** 2
**Overall Recommendation:** 4
**Confidence:** 3

**Summary:**

This paper proposes a method to automatically generate concepts using existing generative models and introduces a validation method. When generating concepts, it proposes a concept activation vector (CAV) by dividing the dataset into positive and negative samples. Additionally, it proposes a method to eliminate the mask effect of each neural network when extracting the concept (CAV) from an image. From the obtained CAV, it combines techniques such as class activation map (CAM) and layer-wise relevance propagation (LRP) to generate an explanation map that highlights the influential factors in the model's decision. Subsequently, it additionally proposes a validation method based on cosine similarity.

**Compliance With Llm Reviewing Policy:**

Affirmed.

**Final Justification:**

The authors' response has addressed my concerns.

**Key Questions For Authors:**

1. What are the elements of a concept as defined or structured by the authors?
2. Directionality is important in the authors' CAV method. How are negative samples defined in cases of multiple labels or labels that are difficult to distinguish?

**Limitations:**

It shares the same limitations as existing automatic concept methods.

**Strengths And Weaknesses:**

This paper well-summarizes existing methods for automatically generating concepts and achieves high performance by solving problems based on them. As generative models advance, there is an expectation that improved concepts could be generated, and various combinations are anticipated in the generation of explanation maps. However, it retains the inherent limitations of existing methods. In particular, an in-depth consideration of the characteristics that concepts should possess is necessary.

---

> ### Author Rebuttal · Authors · 2026-03-31
>
> We agree that ConEx shares with prior CAV-based work the use of a single vector to represent each concept. However, we highlight three reasons this does not critically limit ConEx: **(1)** segment-level extraction reduces intra-concept variance compared to image-level CAVs (+15% VCM, +13% CCM, Appendix H.3), meaning the positive set is already more homogeneous; **(2)** for concepts with heterogeneous appearances, VCM detects and flags low-quality CAVs rather than silently failing - making this a graceful degradation rather than a silent limitation; **(3)** ConEx even outperforms MCD (Tab.3), which explicitly uses multiple concept directions per concept, suggesting that concept-pure grounding is more effective than representational complexity alone. The limitation of single-vector representation is thus detected and bounded by our validation framework. We will explicitly flag this as a known limitation in Appendix K and note multi-direction extension as a concrete future direction.
>
>
> ## (Q1) “What are the elements of a concept as defined or structured by the authors?”
>
> A concept in ConEx is a four-part object: **(i)** a natural-language phrase (e.g., 'floppy ears'), **(ii)** a set of spatial segments obtained by grounding that phrase in class images via GroundedSAM, **(iii)** a Concept Activation Vector (CAV) encoding its direction in the model's latent space as a centroid difference, and **(iv)** a spatial saliency map via CWV (Eq. 2–4) showing where it activates in a given image. This two-sided guarantee, linguistic generation followed by visual verification, ensures that a concept is neither semantically vacuous nor visually ungrounded.
> Crucially, a candidate phrase only becomes a concept after surviving two empirical quality filters: it must appear in ≥15% of class images (occurrence rate) and its grounded segments must cover ≥20% of the class object region (spatial coverage). These filters operationalize the requirement that a concept must be visually consistent - meaning it can be reliably detected across many instances of the class , and spatially coherent - meaning it corresponds to a meaningful portion of the object, not a background artifact. Concepts that survive these filters achieve an average human-rated correspondence score of 8.8/10 (Appendix G.3), confirming that our automatically derived definition aligns with human judgment. We will add a 'concept definition' at the beginning of Section 3 to make this explicit.
> In summary, **a concept in ConEx satisfies**: (1) linguistic interpretability, (2) visual consistency, (3) spatial coherence, and (4) latent-space discriminability.
>
>
> ## (Q2) “Directionality is important in the authors' CAV method. How are negative samples defined in cases of multiple labels or labels that are difficult to distinguish?”
> Our negative samples are drawn from segments of other concepts from randomly selected classes - not from a complementary class or "not-k" category. We believe this design choice addresses both scenarios the reviewer raises.
>
> For **multi-label settings**, no well-defined class complement exists, so class-complement negatives would be ill-defined. Our approach sidesteps this entirely: the negative set is diverse by construction, requiring only that it provides a stable contrastive anchor in latent space - not that it semantically opposes the positive concept.
>
> For **hard-to-distinguish classes** (e.g., two similar dog breeds), negatives are drawn from many random classes rather than the single most similar one, so the negative centroid is not dominated by any near-duplicate. The centroid-difference formulation (Eq. 1) averages over a broad distribution, making it robust to individual similar classes contaminating the direction - and more robust than SVM-based approaches (Martin & Weller, 2019), whose decision boundary is sensitive to samples near the margin.
>
> For cases where stricter separation is required, Appendix D.1 describes an optional enhancement that explicitly filters negative concepts by requiring low cosine similarity between their mean latent embedding and that of the target concept - guaranteeing that negatives are visually and semantically distinct from the positive concept by construction, at the cost of additional computation.
>
> Finally, VCM directly measures whether the resulting CAV aligns with held-out segments of the target concept, providing an automatic quality flag for any CAV whose directionality is compromised by negative set noise. The high VCM scores across 3,602 IN concepts (0.68 for RN, Tab.3) empirically confirm that our CAVs are reliably directional even for fine-grained concepts.

---

> > ### Author Rebuttal · Reviewer_3RcT · 2026-04-04
> >
> > I have carefully reviewed the paper with a particular focus on the limitations of automatic concept methods. I believe that the limitations of these automatic concept methods should be further discussed in future work, grounded in the explicit definition addressed during this review process. The authors' rebuttal has resolved my concerns, and I will be raising my score accordingly.

---

> > > ### Author Response · Authors · 2026-04-04
> > >
> > > We sincerely thank Reviewer 3RcT for their careful reading of our rebuttal and for the willingness to raise the score. We are glad that our rebuttal addressed the concerns.
> > > We will expand our discussion on the limitations of automatic concept methods in the revised version, grounding it specifically in the explicit definition of a concept. We appreciate the valuable comments and support.

---

### Official Review · Reviewer_ix9z · 2026-03-12

**Soundness:** 3
**Presentation:** 2
**Significance:** 3
**Originality:** 2
**Overall Recommendation:** 4
**Confidence:** 5

**Summary:**

The paper proposes a framework called ConEx by combining existing techniques to generate concept-based explanations in terms of better localized saliency maps. In addition, they propose two evaluation metrics to assess reliability of the generated explanations. The main advantage of ConEx is it can identify the concepts automatically without human intervention. The authors have conducted comprehensive experiments using multiple datasets on multiple image classification models and have compared their approach with existing feature attribution methods and concept-based explanation methods.

**Compliance With Llm Reviewing Policy:**

Affirmed.

**Final Justification:**

While the authors have answered some of my concerns, overall rebuttal (including other reviews and author responses to them) didnt convince me to increase the score. Hence I will keep my original score.

**Key Questions For Authors:**

1. What exactly will be provided as the explanations, only saliency maps? If saliency maps are they will be concept-based or class-based?
2. Why have you excluded the concept-based explanation methods and ViTs architectures from the faithfulness explanations?
3. Why have you excluded ICE from the human evaluation?
4. What is your intuition for the multiplicative fusion in Eq 5?
5. You have conducted experiments using only a sample of instances and concepts. Can you explain in each experiment, how did you select this sample?

**Limitations:**

Currently a discussion of limitations and future work is missing. Please consider adding it.

**Strengths And Weaknesses:**

Soundness: The paper is technically sound. The proposed method, ConEx, has been built by combining existing methods appropriately. The authors have clearly explained and supported their design decisions. The paper presents a comprehensive experimental study to evaluate various aspects of the proposed method across multiple datasets (coarse- and fine-grained) and image classification architectures. The authors have compared their approach with both existing feature attribution methods and concept-based explanation methods. However, experiments have not been conducted consistently. For example, the faithfulness results (Tables 1 and 2) are compared only with feature attribution methods and results for concept-based explanation methods are not reported. Results for ViTs are also not reported in those tables. Also, qualitative results for other concept-based explanation methods are not included. Human evaluation would have been stronger if feature attribution methods were also included.

Presentation: The paper is very well written. I enjoyed reading the paper. The authors have always backed up their design decisions and explained the proposed method well. However, the figures could be made more visible and clearer. Also, it would have been better to include the explanations generated by the proposed method for the judgment of the reader. While some explanations are provided in Figure 3, they are class-level rather than concept-level. Also, I prefer that the results for each experiment are placed in the same section as its description for easy reference.

Significance - The paper addresses an important research question, improving automatic concept-based explanations for image classification models. The experimental results, although not all experiments cover all architectures and methods, indicate that the proposed method outperforms existing work. The proposed method will be useful for practitioners.

Originality - The paper appropriately combines existing methods, and the experimental results show that this combination provides more localised, concept-based explanations. The paper doesn't necessarily work on a problem or introduce a new methodology or data.

---

> ### Author Rebuttal · Authors · 2026-03-31
>
> **We thank Reviewer ix9z for the comprehensive and positive review, and address each point below.**
>
> ### Figures
> We will increase font sizes and improve clarity throughout. We will also add a dedicated figure showing concept-level saliency maps alongside their textual labels and class-level aggregations, and a side-by-side qualitative comparison of ACE, ICE, MCD, and ConEx.
> ### Class- vs. concept-level explanations (Figure 3)
> ConEx produces both: concept-level maps (Eq. 2–4) show where each individual concept (e.g., "yellow beak") activates, while the class-level map (Eq. 7) aggregates these weighted by concept importance. Figure 3 shows the latter; we will add a dedicated panel showing the per-concept maps that feed into it.
> ### Experiment placement
> We will place each results table/figure immediately following its experimental description.
> ### Human evaluation scope
> Feature attribution methods are evaluated in the Agreement & Distinction studies (Table 7, Appendix G.2) - ConEx outperforms them. They are excluded from Understandability because it is concept-based: participants match a highlighted region to one of five predefined semantic concepts. This requires explicit concept grounding, which heatmap-based methods lack, introducing a structural mismatch and confound the evaluation. The protocol follows prior work, ensuring fairness and comparability. We will clarify this.
> ### Limitations are missing
> **Limitations and directions for future work are provided in Appendix K (referred in Lines 437–438).**
>
>
> ## (Q1) What will be provided as the explanations?
>
> ConEx produces both concept-level and class-level explanations. At the concept level, each CAV yields a spatial saliency map (Eq. 2–4) that highlights where a specific concept is recognized by the network and how strongly it activates. At the class level, these are aggregated via concept importance weights (Eq. 6–7) into a single class-specific explanation map that reveals which concepts drove the prediction and where. Figure 1 and Section 3.5 describe this precisely.
>
> ## (Q2) Excluding concept-based methods and ViTs from Tables 1 & 2
> Tables 1&2 report pixel-level perturbation faithfulness (Insertion/Deletion), which requires each method to produce a single pixel-level attribution map per image. ACE, ICE, and MCD do not do this - they produce concept-level importance scores. Naively aggregating their concept maps into a saliency map would require method-specific design choices absent from their original implementations, producing an unfair comparison. We therefore evaluate concept-based methods on their own terms via CINS/CDEL (Table 3), the protocol used in their original papers. **ConEx is the only method evaluated on both axes precisely because it is the only method that produces both pixel-level saliency maps and concept-level representations**. We will add a bridging statement in the paper making this explicit.
> Regarding ViTs: producing spatially localized saliency maps for ViTs is an open problem we do not claim to solve (Appendix E, Sections 2.3 and 3). Our ViT pipeline produces high-quality global concept attributions (Table 3); including ViT in Tables 1&2 would therefore be misleading rather than informative.
>
> ## (Q3) Excluding ICE from the understandability table
> ICE is excluded for two reasons: its non-negative CAV constraint makes concept representations harder to visualize as prototypical patches, complicating direct protocol comparison; and including it would add three pairwise statistical comparisons, significantly expanding the table. This is a presentation decision - ICE was fully evaluated under the identical protocol (SA=61.34, PPR=55.84, INNS=0.42, INTS=0.38), with results consistent with our reported conclusions. We will include the complete ICE results in Appendix G.
>
> ## (Q4) Intuition for multiplicative fusion (Eq.5)
>
> Multiplicative fusion enforces a strict logical AND: attribution is amplified only where both concept presence ($M_k > 0$) and model relevance (LRP score $> 0$) coincide. A region with high model relevance but no concept presence, or vice versa, contributes minimally. This suppresses spurious activations, a known weakness of additive approaches where strong background relevance can propagate into the final map even without a meaningful concept. We will add this explanation.
>
> ## (Q5) Sample selection details
> Full details are in Sections 4.2 and Appendices G and J; we will consolidate them into a dedicated paragraph in Appendix J. Briefly: (a) Faithfulness (Tables 1, 2): full validation splits of CUB/IN/SD; FunnyBirds uses 500 images × 50 classes with predefined parts per the benchmark. (b) Concept quality (Table 3): 100 IN classes × 1,000 images; VCM uses a 20% held-out set (200 concepts, 25 segments each); CCM uses 50 classes × 5 concepts × 20 segments = 5,000 samples. (c) Human study: 58 participants, 10 random samples per class/method, 300 distinct samples total, uniquely randomized per participant.

---

> > ### Author Rebuttal · Reviewer_ix9z · 2026-04-02
> >
> > I thank the authors for replying to my comments. However, I do not fully agree with the authors' reply to Q2; you can adapt concept-based methods for use in faithfulness experiments. Also, as indicated by other reviewers, the paper doesn't provide any methodological innovation but a combination of existing methods. Hence, I will not change my rating.

---

> > > ### Author Response · Authors · 2026-04-03
> > >
> > > We sincerely thank the reviewer for their continued engagement and their positive overall assessment. We address the two remaining points.
> > >
> > > ## On adapting concept-based methods for pixel-level faithfulness (Q2)
> > > We did attempt to adapt ACE, ICE, and MCD for pixel-level evaluation. The difficulty is that doing so requires design choices - thresholding, normalization, and spatial back-projection of concept scores - that are absent from the original implementations and introduce confounds unrelated to each method's core contribution. Any such adapted variant would therefore be an untested system that the original authors neither designed nor validated, making a fair comparison difficult.
> > > We therefore follow a principled protocol: evaluating each method in the setting it was designed for (INS/DEL for saliency methods, CINS/CDEL for concept-based methods), and complement this with VCM/CCM for additional coverage.
> > > To further strengthen the paper, we will include an adapted pixel-level comparison in the Appendix with explicit caveats regarding these design choices.
> > >
> > > ## On methodological innovation
> > > We appreciate the reviewer's perspective. ConEx’s central contribution is the first fully automatic framework that simultaneously discovers, spatially grounds, and validates class-specific concepts, producing explanations that are both semantically grounded at the concept level and spatially faithful at the pixel level - a capability no prior saliency or concept-based method achieves in isolation.
> > >
> > > The non-obvious design choices that make this possible include architecture-specific embedding strategies that preserve semantic purity (e.g., using layer-wise masking with neighborhood padding), centroid-difference CAV construction from precisely grounded concept segments that outperforms SVM-based and image-level alternatives (Tables 3, 8, Appendix H.3), channel-weighted vectors for position-independent concept localization, and multiplicative fusion with LRP - whose conservation property ensures fully accounted relevance across layers. The VCM and CCM metrics further enable quantitative concept validation without manual annotation for the first time.
> > >
> > > The contribution therefore lies in what the full framework creates - a fundamentally new capability: **faithful attribution using spatially localized concepts**. This is **consistently reflected by state-of-the-art results across faithfulness, concept quality, segmentation, and human interpretability benchmarks**.
> > >
> > > We believe these capabilities make ConEx a highly practical and rigorous framework for the interpretability community, and would greatly appreciate the reviewer's support for its acceptance.
> > >
> > > ## Summary of changes we will make:
> > > - Add a dedicated figure showing per-concept saliency maps with textual labels, plus a side-by-side qualitative comparison with ACE, ICE, and MCD.
> > > - Improve clarity and font sizes in all figures.
> > > - Move the Limitations and Future Work section from Appendix K to the main paper.
> > > - Add a clarifying note in Section 4.2.1 explaining the evaluation protocol for concept-based vs. pixel-based methods.
> > > - Include an adapted pixel-level comparison in the Appendix with appropriate caveats.

---

### Decision · Program_Chairs · 2026-04-30

**Decision:**

Accept (regular)

**Comment:**

This paper proposes a post-hoc framework for automatic concept discovery, concept localization, and class-specific attribution in image classifiers. Overall, the problem is important, and the paper is well written and organized.

However, the major concern lies in the methodological novelty, which is essentially in determining the acceptance or not. Reviewer ix9z and especially GGjS questioned whether the contribution goes sufficiently beyond a composition of existing components, while Reviewer uysE raised concerns about the pipeline nature of the method and some evaluation choices. At the same time, some of the strongest negative claims in GGjS rely on factual inaccuracies, including that ViT results are absent and that ConEx is essentially a CBM-style method; neither is correct, which lowers the weight AC places on that review.

Overall, the AC believe that this is strong borderline paper, on the basis that not clearly meeting the ICML bar for methodological novelty despite being a solid and interesting piece of work. However, the AC also thinks the paper offers a useful and well-validated integrated capability that is not cleanly provided by prior saliency or concept-based methods alone, and therefore recommend weak accept, with low priority if space in the program is limited. If accepted, the final version should better calibrate the novelty claims and more clearly discuss the method’s limitations and scope.